# FASTSHAP: REAL-TIME SHAPLEY VALUE ESTIMATION

**Neil Jethani**[*]
New York University

**Mukund Sudarshan**[*]
New York University

**Ian Covert**[*]
University of Washington

**Su-In Lee**
University of Washington

**Rajesh Ranganath**
New York University

## ABSTRACT

Although Shapley values are theoretically appealing for explaining black-box models, they are costly to calculate and thus impractical in settings that involve large, high-dimensional models. To remedy this issue, we introduce FastSHAP, a new method for estimating Shapley values in a single forward pass using a learned explainer model. To enable efficient training without requiring ground truth Shapley values, we develop an approach to train FastSHAP via stochastic gradient descent using a weighted least squares objective function. In our experiments with tabular and image datasets, we compare FastSHAP to existing estimation approaches and find that it generates accurate explanations with an orders-of-magnitude speedup.

## 1 INTRODUCTION

With the proliferation of black-box models, Shapley values (Shapley, 1953) have emerged as a popular explanation approach due to their strong theoretical properties (Lipovetsky and Conklin, 2001; Štrumbelj and Kononenko, 2014; Datta et al., 2016; Lundberg and Lee, 2017). In practice, however, Shapley value-based explanations are known to have high computational complexity, with an exact calculation requiring an exponential number of model evaluations (Van den Broeck et al., 2021). Speed becomes a critical issue as models increase in size and dimensionality, and for the largest models in fields such as computer vision and natural language processing, there is an unmet need for significantly faster Shapley value approximations that maintain high accuracy.

Recent work has addressed the computational challenges with Shapley values using two main approaches. First, many works have proposed *stochastic estimators* (Castro et al., 2009; Štrumbelj and Kononenko, 2014; Lundberg and Lee, 2017; Covert et al., 2020) that rely on sampling either feature subsets or permutations; though often consistent, these estimators require many model evaluations and impose an undesirable trade-off between run-time and accuracy. Second, some works have proposed *model-specific approximations*, e.g., for trees (Lundberg et al., 2020) or neural networks (Shrikumar et al., 2017; Chen et al., 2018b; Ancona et al., 2019; Wang et al., 2021); while generally faster, these approaches can still require many model evaluations, often induce bias, and typically lack flexibility regarding the handling held-out features—a subject of ongoing debate in the field (Aas et al., 2019; Janzing et al., 2020; Frye et al., 2020; Covert et al., 2021).

Here, we introduce a new approach for efficient Shapley value estimation: to achieve the fastest possible run-time, we propose learning a separate explainer model that outputs precise Shapley value estimates in a single forward pass. Naïvely, such a learning-based approach would seem to require a large training set of ground truth Shapley values, which would be computationally intractable. Instead, our approach trains an explainer model by minimizing an objective function inspired by the Shapley value's weighted least squares characterization (Charnes et al., 1988), which enables efficient gradient-based optimization.

**Our contributions.** We introduce FastSHAP, an amortized approach for generating real-time Shapley value explanations.[1] We derive an objective function from the Shapley value's weighted least

---

[*]Equal contribution
[1]https://git.io/JCqFV (PyTorch), https://git.io/JCqbP (TensorFlow)

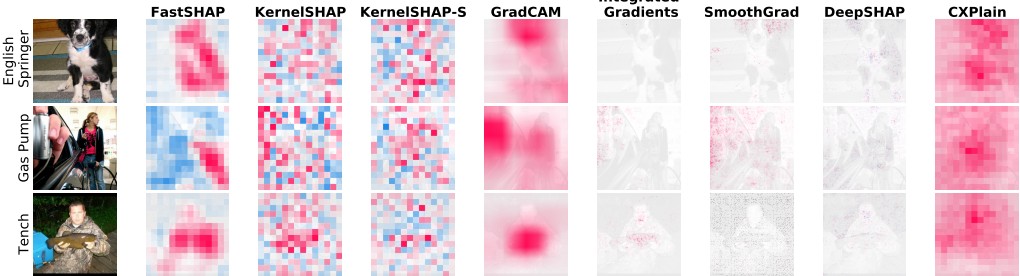

Figure 1: **Explanations generated by each method for Imagenette images.**

squares characterization and investigate several ways to reduce gradient variance during training. Our experiments show that FastSHAP provides accurate Shapley value estimates with an orders-of-magnitude speedup relative to non-amortized estimation approaches. Finally, we also find that FastSHAP generates high-quality image explanations (fig. 1) that outperform gradient-based methods (e.g., IntGrad and GradCAM) on quantitative inclusion and exclusion metrics.

## 2 BACKGROUND

In this section, we introduce notation used throughout the paper and provide an overview of Shapley values and their weighted least squares characterization. Let $\mathbf{x} \in \mathcal{X}$ be a random vector consisting of $d$ features, or $\mathbf{x} = (\mathbf{x}_1, \ldots, \mathbf{x}_d)$, and let $\mathbf{y} \in \mathcal{Y} = \{1, \ldots, K\}$ be the response variable for a classification problem. We use $\mathbf{s} \in \{0,1\}^d$ to denote subsets of the indices $\{1, \ldots, d\}$ and define $\mathbf{x}_s := \{\mathbf{x}_i\}_{i:s_i=1}$. The symbols $\mathbf{x}, \mathbf{y}, \mathbf{s}$ are random variables and $x, y, s$ denote possible values. We use $\mathbf{1}$ and $\mathbf{0}$ to denote vectors of ones and zeros in $\mathbb{R}^d$, so that $\mathbf{1}^\top s$ is a subset's cardinality, and we use $e_i$ to denote the $i$th standard basis vector. Finally, $f(\mathbf{x}; \eta) : \mathcal{X} \mapsto \Delta^{K-1}$ is a model that outputs a probability distribution over $\mathbf{y}$ given $\mathbf{x}$, and $f_y(\mathbf{x}; \eta)$ is the probability for the $y$th class.

### 2.1 SHAPLEY VALUES

Shapley values were originally developed as a credit allocation technique in cooperative game theory (Shapley, 1953), but they have since been adopted to explain predictions from black-box machine learning models (Štrumbelj and Kononenko, 2014; Datta et al., 2016; Lundberg and Lee, 2017). For any value function (or set function) $v : 2^d \mapsto \mathbb{R}$, the Shapley values $\phi(v) \in \mathbb{R}^d$, or $\phi_i(v) \in \mathbb{R}$ for each feature $i = 1, \ldots, d$, are given by the formula

$$\phi_i(v) = \frac{1}{d} \sum_{s_i \neq 1} \binom{d-1}{\mathbf{1}^\top s}^{-1} \left( v(s + e_i) - v(s) \right). \tag{1}$$

The difference $v(s + e_i) - v(s)$ represents the $i$th feature's contribution to the subset $s$, and the summation represents a weighted average across all subsets that do not include $i$. In the model explanation context, the value function is chosen to represent how an individual prediction varies as different subsets of features are removed. For example, given an input-output pair $(x, y)$, the prediction for the $y$th class can be represented by a value function $v_{x,y}$ defined as

$$v_{x,y}(s) = \text{link} \left( \underset{p(\mathbf{x}_{1-s})}{\mathbb{E}} \left[ f_y\left(x_s, \mathbf{x}_{1-s}; \eta\right) \right] \right), \tag{2}$$

where the held out features $\mathbf{x}_{1-s}$ are marginalized out using their joint marginal distribution $p(\mathbf{x}_{1-s})$, and a link function (e.g., logit) is applied to the model output. Recent work has debated the properties of different value function formulations, particularly the choice of how to remove features (Aas et al., 2019; Janzing et al., 2020; Frye et al., 2020; Covert et al., 2021). However, regardless of the formulation, this approach to model explanation enjoys several useful theoretical properties due to its use of Shapley values: for example, the attributions are zero for irrelevant features, and they are guaranteed to sum to the model's prediction. We direct readers to prior work for a detailed discussion of these properties (Lundberg and Lee, 2017; Covert et al., 2021).

Unfortunately, Shapley values also introduce computational challenges: the summation in eq. (1) involves an exponential number of subsets, which makes it infeasible to calculate for large $d$. Fast approximations are therefore required in practice, as we discuss next.

## 2.2 KERNELSHAP

KernelSHAP (Lundberg and Lee, 2017) is a popular Shapley value implementation that relies on an alternative Shapley value interpretation. Given a value function $v_{x,y}(\mathbf{s})$, eq. (1) shows that the values $\phi(v_{x,y})$ are the features' weighted average contributions; equivalently, their weighted least squares characterization says that they are the solution to an optimization problem over $\phi_{x,y} \in \mathbb{R}^d$,

$$\phi(v_{x,y}) = \underset{\phi_{x,y}}{\arg\min} \; \underset{p(\mathbf{s})}{\mathbb{E}} \left[ \left( v_{x,y}(\mathbf{s}) - v_{x,y}(\mathbf{0}) - \mathbf{s}^\top \phi_{x,y} \right)^2 \right] \tag{3}$$

$$\text{s.t.} \quad \mathbf{1}^\top \phi_{x,y} = v_{x,y}(\mathbf{1}) - v_{x,y}(\mathbf{0}), \quad \text{(Efficiency constraint)}$$

where the distribution $p(\mathbf{s})$ is defined as

$$p(s) \propto \frac{d-1}{\binom{d}{\mathbf{1}^\top s} \cdot \mathbf{1}^\top s \cdot (d - \mathbf{1}^\top s)} \quad \text{(Shapley kernel)}$$

for $s$ such that $0 < \mathbf{1}^\top s < d$ (Charnes et al., 1988). Based on this view of the Shapley value, Lundberg and Lee (2017) introduced KernelSHAP, a stochastic estimator that solves an approximate version of eq. (3) given some number of subsets sampled from $p(\mathbf{s})$. Although the estimator is consistent and empirically unbiased (Covert and Lee, 2021), KernelSHAP often requires many samples to achieve an accurate estimate, and it must solve eq. (3) separately for each input-output pair $(x, y)$. As a result, it is unacceptably slow for some use cases, particularly in settings with large, high-dimensional models. Our approach builds on KernelSHAP, leveraging the Shapley value's weighted least squares characterization to design a faster, amortized estimation approach.

## 3 FASTSHAP

We now introduce FastSHAP, a method that amortizes the cost of generating Shapley values across many data samples. FastSHAP has two main advantages over existing approaches: (1) it avoids solving separate optimization problems for each input to be explained, and (2) it can use similar data points to efficiently learn the Shapley value function $\phi(v_{x,y})$.

### 3.1 AMORTIZING SHAPLEY VALUES

In our approach, we propose generating Shapley value explanations using a learned parametric function $\phi_{\text{fast}}(\mathbf{x}, \mathbf{y}; \theta) : \mathcal{X} \times \mathcal{Y} \mapsto \mathbb{R}^d$. Once trained, the parametric function can generate explanations in a single forward pass, providing a significant speedup over methods that approximate Shapley values separately for each sample $(x, y)$. Rather than using a dataset of ground truth Shapley values for training, we train $\phi_{\text{fast}}(\mathbf{x}, \mathbf{y}; \theta)$ by penalizing its predictions according to the weighted least squares objective in eq. (3), or by minimizing the following loss,

$$\mathcal{L}(\theta) = \underset{p(\mathbf{x})}{\mathbb{E}} \underset{\text{Unif}(\mathbf{y})}{\mathbb{E}} \underset{p(\mathbf{s})}{\mathbb{E}} \left[ \left( v_{\mathbf{x},\mathbf{y}}(\mathbf{s}) - v_{\mathbf{x},\mathbf{y}}(\mathbf{0}) - \mathbf{s}^\top \phi_{\text{fast}}(\mathbf{x}, \mathbf{y}; \theta) \right)^2 \right], \tag{4}$$

where $\text{Unif}(\mathbf{y})$ represents a uniform distribution over classes. If the model's predictions are forced to satisfy the Efficiency constraint, then given a large enough dataset and a sufficiently expressive model class for $\phi_{\text{fast}}$, the global optimizer $\phi_{\text{fast}}(\mathbf{x}, \mathbf{y}; \theta^*)$ is a function that outputs exact Shapley values (see proof in appendix A). Formally, the global optimizer satisfies the following:

$$\phi_{\text{fast}}(\mathbf{x}, \mathbf{y}; \theta^*) = \phi(v_{\mathbf{x},\mathbf{y}}) \text{ almost surely in } p(\mathbf{x}, \mathbf{y}). \tag{5}$$

We explore two approaches to address the efficiency requirement. First, we can enforce efficiency by adjusting the Shapley value predictions using their *additive efficient normalization* (Ruiz et al., 1998), which applies the following operation to the model's outputs:

$$\phi_{\text{fast}}^{\text{eff}}(x, y; \theta) = \phi_{\text{fast}}(x, y; \theta) + \frac{1}{d} \underbrace{\left( v_{x,y}(\mathbf{1}) - v_{x,y}(\mathbf{0}) - \mathbf{1}^\top \phi_{\text{fast}}(x, y; \theta) \right)}_{\text{Efficiency gap}}. \tag{6}$$

The normalization step can be applied at inference time and optionally during training; in appendix B, we show that this step is guaranteed to make the estimates closer to the true Shapley values. Second, we can relax the efficiency property by augmenting $\mathcal{L}(\theta)$ with a penalty on the efficiency gap (see eq. (6)); the penalty requires a parameter $\gamma > 0$, and as we set $\gamma \to \infty$ we can guarantee that efficiency holds (see appendix A). Algorithm 1 summarizes our training approach.

**Empirical considerations.** Optimizing $\mathcal{L}(\theta)$ using a single set of samples $(x, y, s)$ is problematic because of high variance in the gradients, which can lead to poor optimization. We therefore consider several steps to reduce gradient variance. First, as is conventional in deep learning, we minibatch across multiple samples from $p(\mathbf{x})$. Next, when possible, we calculate the loss jointly across all classes $y \in \{1, \ldots, K\}$. Then, we experiment with using multiple samples $s \sim p(\mathbf{s})$ for each input sample $x$. Finally, we explore *paired sampling*, where each sample $s$ is paired with its complement $\mathbf{1} - s$, which has been shown to reduce KernelSHAP's variance (Covert and Lee, 2021). Appendix C shows proofs that these steps are guaranteed to reduce gradient variance, and ablation experiments in appendix D demonstrate their improvement on FastSHAP's accuracy.

### 3.2 A DEFAULT VALUE FUNCTION FOR FASTSHAP

FastSHAP has the flexibility to work with any value function $v_{x,y}(\mathbf{s})$. Here, we describe a default value function that is useful for explaining predictions from a classification model.

The value function's aim is to assess, for each subset $s$, the classification probability when only the features $\mathbf{x}_s$ are observed. Because most models $f(\mathbf{x}; \eta)$ do not support making predictions without all the features, we require an approximation that simulates the inclusion of only $\mathbf{x}_s$ (Covert et al., 2021). To this end, we use a supervised surrogate model (Frye et al., 2020; Jethani et al., 2021) to approximate marginalizing out the remaining features $\mathbf{x}_{\mathbf{1}-s}$ using their conditional distribution.

---

**Algorithm 1:** FastSHAP training

**Input:** Value function $v_{x,y}$, learning rate $\alpha$
**Output:** FastSHAP explainer $\phi_{\text{fast}}(\mathbf{x}, \mathbf{y}; \theta)$
initialize $\phi_{\text{fast}}(\mathbf{x}, \mathbf{y}; \theta)$
**while** *not converged* **do**
    sample $x \sim p(\mathbf{x})$, $y \sim \text{Unif}(\mathbf{y})$, $s \sim p(\mathbf{s})$
    predict $\hat{\phi} \leftarrow \phi_{\text{fast}}(x, y; \theta)$
    **if** *normalize* **then**
        set $\hat{\phi} \leftarrow$
        $\hat{\phi} + d^{-1} \left( v_{x,y}(\mathbf{1}) - v_{x,y}(\mathbf{0}) - \mathbf{1}^T \hat{\phi} \right)$
    **end**
    calculate
    $\mathcal{L} \leftarrow \left( v_{x,y}(s) - v_{x,y}(\mathbf{0}) - s^T \hat{\phi} \right)^2$
    update $\theta \leftarrow \theta - \alpha \nabla_\theta \mathcal{L}$
**end**

---

Separate from the original model $f(\mathbf{x}; \eta)$, the *surrogate* model $p_{\text{surr}}(\mathbf{y} \mid m(\mathbf{x}, \mathbf{s}); \beta)$ takes as input a vector of masked features $m(x, s)$, where the masking function $m$ replaces features $x_i$ such that $s_i = 0$ with a `[mask]` value that is not in the support of $\mathcal{X}$. Similar to prior work (Frye et al., 2020; Jethani et al., 2021), the parameters $\beta$ are learned by minimizing the following loss function:

$$\mathcal{L}(\beta) = \underset{p(\mathbf{x})}{\mathbb{E}} \underset{p(\mathbf{s})}{\mathbb{E}} \left[ D_{\text{KL}} \big( f(\mathbf{x}; \eta) \,||\, p_{\text{surr}}(\mathbf{y} \mid m(\mathbf{x}, \mathbf{s}); \beta) \big) \right]. \tag{7}$$

It has been shown that the global optimizer to eq. (7), or $p_{\text{surr}}(\mathbf{y} \mid m(\mathbf{x}, \mathbf{s}); \beta^*)$, is equivalent to marginalizing out features from $f(\mathbf{x}; \eta)$ with their conditional distribution (Covert et al., 2021):

$$p_{\text{surr}}(y \mid m(x, s); \beta^*) = \mathbb{E}[f_y(\mathbf{x}; \eta) \mid \mathbf{x}_s = x_s]. \tag{8}$$

The choice of distribution over $p(\mathbf{s})$ does not affect the global optimizer of eq. (7), but we use the Shapley kernel to put more weight on subsets likely to be encountered when training FastSHAP. We use the surrogate model as a default choice for two reasons. First, it requires a single prediction for each evaluation of $v_{x,y}(s)$, which permits faster training than the common approach of averaging across many background samples (Lundberg and Lee, 2017; Janzing et al., 2020). Second, it yields explanations that reflect the model's dependence on the *information* communicated by each feature, rather than its *algebraic* dependence (Frye et al., 2020; Covert et al., 2021).

## 4 RELATED WORK

Recent work on Shapley value explanations has largely focused on how to remove features (Aas et al., 2019; Frye et al., 2020; Covert et al., 2021) and how to approximate Shapley values efficiently (Chen et al., 2018b; Ancona et al., 2019; Lundberg et al., 2020; Covert and Lee, 2021). Model-specific approximations are relatively fast, but they often introduce bias and are entangled with specific feature removal approaches (Shrikumar et al., 2017; Ancona et al., 2019; Lundberg et al., 2020). In contrast, model-agnostic stochastic approximations are more flexible, but they must trade off run-time and accuracy in the explanation. For example, KernelSHAP samples subsets to approximate the solution to a weighted least squares problem (Lundberg and Lee, 2017), while other

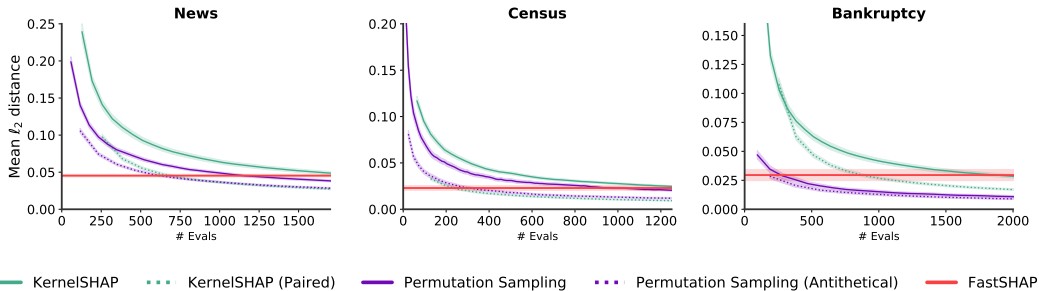

Figure 2: **Comparison of Shapley value approximation accuracy across methods.** Using three datasets, we measure the distance of each method's estimates to the ground truth as a function of the number of model evaluations. FastSHAP is represented by a horizontal line since it requires only a single forward pass. The baselines require $200-2000\times$ model evaluations to achieve FastSHAP's level of accuracy.

approaches sample marginal contributions (Castro et al., 2009; Štrumbelj and Kononenko, 2014) or feature permutations (Illés and Kerényi, 2019; Mitchell et al., 2021). FastSHAP trains an explainer model to output an estimate that would otherwise require orders of magnitude more model evaluations, and, unlike other fast approximations, it is agnostic to the model class and feature removal approach.

Other methods have been proposed to generate explanations using learned explainer models. These are referred to as *amortized explanation methods* (Covert et al., 2021; Jethani et al., 2021), and they include several approaches that are comparable to gradient-based methods in terms of compute time (Dabkowski and Gal, 2017; Chen et al., 2018a; Yoon et al., 2018; Schwab and Karlen, 2019; Schulz et al., 2020; Jethani et al., 2021). Notably, one approach generates a training dataset of ground truth explanations and then learns an explainer model to output explanations directly (Schwab and Karlen, 2019)—a principle that can be applied with any attribution method, at least in theory. However, for Shapley values, generating a large training set would be very costly, so FastSHAP sidesteps the need for a training set using a custom loss function based on the Shapley value's weighted least squares characterization (Charnes et al., 1988).

## 5   STRUCTURED DATA EXPERIMENTS

We analyze FastSHAP's performance by comparing it to several well-understood baselines. First, we evaluate its accuracy on tabular (structured) datasets by comparing its outputs to the ground truth Shapley values. Then, to disentangle the benefits of amortization from the in-distribution value function, we make the same comparisons using different value function formulations $v_{x,y}(\mathbf{s})$. Unless otherwise stated, we use the surrogate model value function introduced in section 3.2. Later, in section 6, we test FastSHAP's ability to generate image explanations.

**Baseline methods.**   To contextualize FastSHAP's accuracy, we compare it to several non-amortized stochastic estimators. First, we compare to KernelSHAP (Lundberg and Lee, 2017) and its acceleration that uses paired sampling (Covert and Lee, 2021). Next, we compare to a permutation sampling approach and its acceleration that uses antithetical sampling (Mitchell et al., 2021). As a performance metric, we calculate the proximity to Shapley values that were obtained by running KernelSHAP to convergence; we use these values as our ground truth because KernelSHAP is known to converge to the true Shapley values given infinite samples (Covert and Lee, 2021). These baselines were all run using an open-source implementation.[2]

**Implementation details.**   We use either neural networks or tree-based models for each of $f(\mathbf{x}; \eta)$ and $p_{\text{surr}}(\mathbf{y} \mid m(\mathbf{x}, \mathbf{s}); \beta)$. The FastSHAP explainer model $\phi_{\text{fast}}(\mathbf{x}, \mathbf{y}; \theta)$ is implemented with a network $g(\mathbf{x}; \theta) : \mathcal{X} \to \mathbb{R}^d \times \mathcal{Y}$ that outputs a vector of Shapley values for every $y \in \mathcal{Y}$; deep neural networks are ideal for FastSHAP because they have high representation capacity, they can provide many-to-many mappings, and they can be trained by stochastic gradient descent. Appendix D contains more details about our implementation, including model classes, network architectures and training hyperparameters.

---

[2]https://github.com/iancovert/shapley-regression/ (License: MIT)

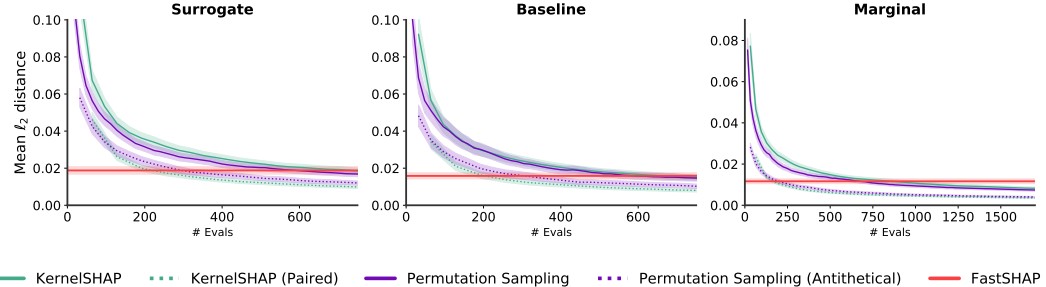

Figure 3: **FastSHAP approximation accuracy for different value functions.** Using the `marketing` dataset, we find that FastSHAP provides accurate Shapley value estimates regardless of the value function (surrogate, marginal, baseline), with the baselines requiring $200-1000\times$ model evaluations to achieve Fast-SHAP's level of accuracy. Error bars represent 95% confidence intervals.

We also perform a series of experiments to determine several training hyperparameters for FastSHAP, exploring (1) whether or not to use paired sampling, (2) the number of subset samples to use, and (3) how to best enforce the efficiency constraint. Based on the results (see appendix D), we use the following settings for our tabular data experiments: we use paired sampling, between 32 and 64 samples of $\mathbf{s}$ per $\mathbf{x}$ sample, additive efficient normalization during both training and inference, and we set $\gamma = 0$ (since the normalization step is sufficient to enforce efficiency).

### 5.1 ACCURACY OF FASTSHAP EXPLANATIONS

Here, we test whether FastSHAP's estimates are close to the ground truth Shapley values. Our experiments use data from a 1994 United States `census`, a bank `marketing` campaign, `bankruptcy` statistics, and online `news` articles (Dua and Graff, 2017). The `census` data contains 12 input features, and the binary label indicates whether a person makes over $50K a year (Kohavi et al., 1996). The `marketing` dataset contains 17 input features, and the label indicates whether the customer subscribed to a term deposit (Moro et al., 2014). The `bankruptcy` dataset contains 96 features describing various companies and whether they went bankrupt (Liang et al., 2016). The `news` dataset contains 60 numerical features about articles published on Mashable, and our label indicates whether the share count exceeds the median number (1400) (Fernandes et al., 2015). The datasets were each split 80/10/10 for training, validation and testing.

In fig. 2, we show the distance of each method's estimates to the ground truth as a function of the number of model evaluations for the `news`, `census` and `bankruptcy` datasets. Figure 3 shows results for the `marketing` dataset with three different value functions (see section 5.2). For the baselines, each sample $\mathbf{s}$ requires evaluating the model given a subset of features, but since Fast-SHAP requires only a single forward pass of $\phi_{\text{fast}}(\mathbf{x}, \mathbf{y}; \theta)$, we show it as a horizontal line.

To reach FastSHAP's level of accuracy on the `news`, `census` and `bankruptcy` datasets, KernelSHAP requires between 1,200-2,000 model evaluations; like prior work (Covert and Lee, 2021), we find that paired sampling improves KernelSHAP's rate of convergence, helping reach Fast-SHAP's accuracy in 250-1,000 model evaluations. The permutation sampling baselines tend to be faster: the original version requires between 300-1,000 evaluations, and antithetical sampling takes 200-500 evaluations to reach an accuracy equivalent to FastSHAP. Across all four datasets, however, FastSHAP achieves its level of accuracy at least at least $600\times$ faster than the original version of KernelSHAP, and $200\times$ faster than the best non-amortized baseline.

### 5.2 DISENTANGLING AMORTIZATION AND THE CHOICE OF VALUE FUNCTION

In this experiment, we verify that FastSHAP produces accurate Shapley value estimates regardless of the choice of value function. We use the `marketing` dataset for this experiment and test the following value functions:

1. (Surrogate/In-distribution) $v_{x,y}(s) = p_{\text{surr}}(y \mid m(x,s); \beta)$

2. (Marginal/Out-of-distribution) $v_{x,y}(s) = \mathbb{E}_{p(\mathbf{x}_{1-s})}\left[f_y\left(x_s, \mathbf{x}_{1-s}; \eta\right)\right]$

3. (Baseline removal) $v_{x,y}(s) = f_y(x_s, x_{1-s}^b; \eta)$, where $x^b \in \mathcal{X}$ are fixed baseline values (the mean for continuous features and mode for discrete ones)

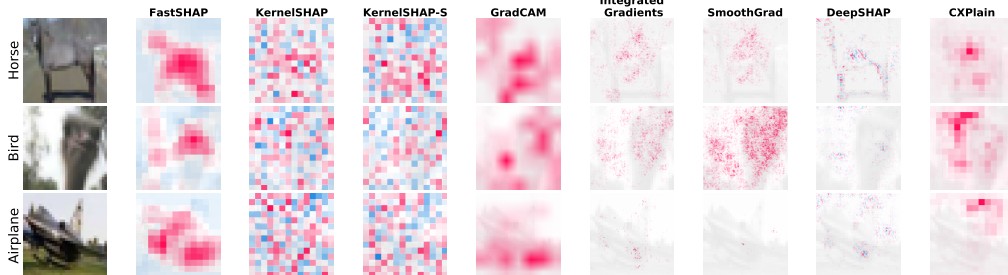

Figure 4: **Explanations generated by each method for CIFAR-10 images.**

In fig. 3 we compare FastSHAP to the same non-amortized baseline methods, where each method generates Shapley value estimates using the value functions listed above. The results show that FastSHAP maintains the same computational advantage across all three cases: to achieve the same accuracy as FastSHAP's single forward pass, the baseline methods require at least 200 model evaluations, but in some cases up to nearly 1,000.

## 6 IMAGE EXPERIMENTS

Images represent a challenging setting for Shapley values due to their high dimensionality and the computational cost of model evaluation. We therefore compare FastSHAP to KernelSHAP on two image datasets. We also consider several widely used gradient-based explanation methods, because they are the most commonly used methods for explaining image classifiers.

### 6.1 DATASETS

We consider two popular image datasets for our experiments. **CIFAR-10** (Krizhevsky et al., 2009) contains 60,000 $32 \times 32$ images across 10 classes, and we use 50,000 samples for training and 5,000 samples each for validation and testing. Each image is resized to $224 \times 224$ using bilinear interpolation to interface with the ResNet-50 architecture (He et al., 2016). Figure 4 shows example CIFAR-10 explanations generated by each method. The **Imagenette** dataset (Howard and Gugger, 2020), a subset of 10 classes from the ImageNet dataset, contains 13,394 total images. Each image is cropped to keep the $224 \times 224$ central region, and the data is split 9,469/1,963/1,962. Example Imagenette explanations are shown in fig. 1.

### 6.2 EXPLANATION METHODS

We test three Shapley value estimators, FastSHAP, KernelSHAP, and DeepSHAP (Lundberg and Lee, 2017), where the last is an existing approximation designed for neural networks. We test KernelSHAP with the zeros baseline value function, which we refer to simply as KernelSHAP, and with the in-distribution surrogate value function, which we refer to as KernelSHAP-S. We also compare these methods to the gradient-based explanation methods GradCAM (Selvaraju et al., 2017), SmoothGrad (Smilkov et al., 2017) and IntGrad (Sundararajan et al., 2017). Gradient-based methods are relatively fast and have therefore been widely adopted for explaining image classifiers. Finally, we also compare to CXPlain (Schwab and Karlen, 2019), an amortized explanation method that generates attributions that are not based on Shapley values.

**Implementation details.** The models $f(\mathbf{x}; \eta)$ and $p_{\text{surr}}(\mathbf{y} \mid m(\mathbf{x}, \mathbf{s}); \beta)$ are both ResNet-50 networks (He et al., 2016) pretrained on ImageNet and fine-tuned on the corresponding imaging dataset. FastSHAP, CXPlain, and KernelSHAP are all implemented to output $14 \times 14$ superpixel attributions for each class. For FastSHAP, we parameterize $\phi_{\text{fast}}(\mathbf{x}, \mathbf{y}; \theta)$ to output superpixel attributions: we use an identical pretrained ResNet-50 but replace the final layers with a $1 \times 1$ convolutional layer so that the output is $14 \times 14 \times K$ (see details appendix D). We use an identical network to produce attributions for CXPlain. For FastSHAP, we do not use additive efficient normalization, and we set $\gamma = 0$; we find that this relaxation of the Shapley value's efficiency property does not inhibit FastSHAP's ability to produce high-quality image explanations. KernelSHAP and KernelSHAP-S are implemented using the `shap`[3] package's default parameters, and GradCAM, SmoothGrad, and IntGrad are implemented using the `tf-explain`[4] package's default parameters.

---

[3] https://shap.readthedocs.io/en/latest/ (License: MIT)
[4] https://tf-explain.readthedocs.io/en/latest/ (License: MIT)

### 6.3 QUALITATIVE REMARKS

Explanations generated by each method are shown in fig. 4 for CIFAR-10 and fig. 1 for Imagenette (see appendix E for more examples). While a qualitative evaluation is insufficient to draw conclusions about each method, we offer several remarks on these examples. FastSHAP, and to some extent GradCAM, appear to reliably highlight the important objects, while the KernelSHAP explanations are noisy and fail to localize important regions. To a lesser extent, CXPlain occasionally highlights important regions. In comparison, the remaining methods (SmoothGrad, IntGrad and DeepSHAP) are granulated and highlight only small parts of the key objects. Next, we consider quantitative metrics that test these observations more systematically.

### 6.4 QUANTITATIVE EVALUATION

Evaluating the quality of Shapley value estimates requires access to ground truth Shapley values, which is computationally infeasible for images. Instead, we use two metrics that evaluate an explanation's ability to identify informative image regions. These metrics build on several recent proposals (Petsiuk et al., 2018; Hooker et al., 2018; Jethani et al., 2021) and evaluate the model's classification accuracy after including or excluding pixels according to their estimated importance.

Similar to Jethani et al. (2021), we begin by training a single evaluation model $p_{\text{eval}}$ to approximate the $f(\mathbf{x}; \eta)$ model's output given a subset of features; this serves as an alternative to training separate models on each set of features (Hooker et al., 2018) and offers a more realistic option than masking features with zeros (Schwab and Karlen, 2019). This procedure is analogous to the $p_{\text{surr}}$ training procedure in section 3.2, except it sets the subset distribution to $p(\mathbf{s}) = \text{Uniform}(\{0, 1\}^d)$ to ensure all subsets are equally weighted.

Next, we analyze how the model's predictions change as we remove either important or unimportant features according to each explanation. Using a set of 1,000 images, each image is first labeled by the original model $f(\mathbf{x}; \eta)$ using the most likely predicted class. We then use explanations generated by each method to produce feature rankings and compute the top-1 accuracy (a measure of agreement with the original model) as we either include or exclude the most important features, ranging from 0-100%. The area under each curve (AUC) is termed the *Inclusion AUC* or *Exclusion AUC*.

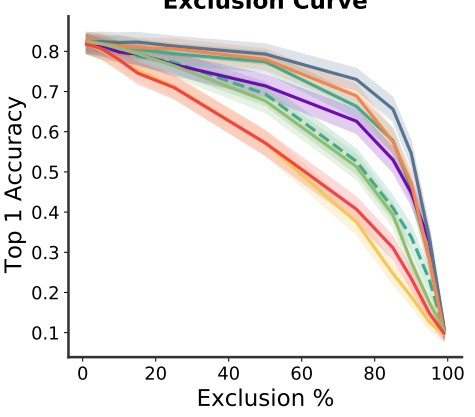

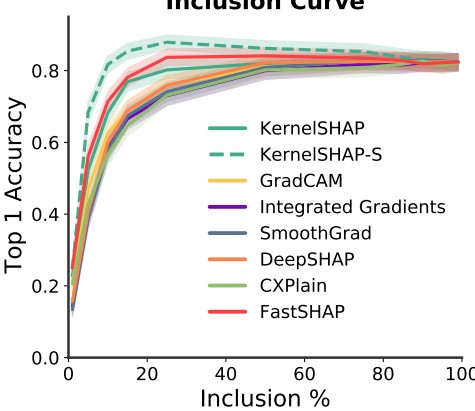

Figure 5: **Imagenette inclusion and exclusion curves.** The change in top-1 accuracy as an increasing percentage of the pixels estimated to be important are excluded (top) or included (bottom).

These metrics match the idea that an accurate image explanation should (1) maximally degrade the performance of $p_{\text{eval}}$ when important features are excluded, and (2) maximally improve the performance of $p_{\text{eval}}$ when important features are included (Petsiuk et al., 2018; Hooker et al., 2018). The explanations are evaluated by removing superpixels; for gradient-based methods, we coarsen the explanations using the sum total importance within each superpixel. In appendix E, we replicate these metrics using log-odds rather than top-1 accuracy, finding a similar ordering among methods.

**Results.** Table 1 shows the Inclusion and Exclusion AUC achieved by each method for both CIFAR-10 and Imagenette. In fig. 5, we also present the curves used to generate these AUCs for Imagenette. Lower Exclusion AUCs and higher Inclusion AUCs are better. These results show that FastSHAP outperforms all baseline methods when evaluated with Exclusion AUC: when the pixels identified as important by FastSHAP are removed from the images, the sharpest decline in top-1 accuracy is observed. Additionally, FastSHAP performs well when evaluated on the basis of Inclusion AUC, second only to KernelSHAP-S.

Table 1: **Exclusion and Inclusion AUCs.** Evaluation of each method on the basis of Exclusion AUC (lower is better) and Inclusion AUC (higher is better) calculated using top-1 accuracy. Parentheses indicate 95% confidence intervals, and the best methods are bolded in each column.

| | CIFAR-10 | | Imagenette | |
|---|---|---|---|---|
| | Exclusion AUC | Inclusion AUC | Exclusion AUC | Inclusion AUC |
| **FastSHAP** | **0.42 (0.41, 0.43)** | 0.78 (0.77, 0.79) | **0.51 (0.49, 0.52)** | **0.79 (0.78, 0.80)** |
| KernelSHAP | 0.64 (0.63, 0.65) | 0.78 (0.77, 0.79) | 0.68 (0.67, 0.70) | 0.77 (0.75, 0.78) |
| KernelSHAP-S | 0.54 (0.52, 0.55) | **0.86 (0.85, 0.87)** | 0.61 (0.60, 0.62) | **0.82 (0.80, 0.83)** |
| GradCAM | 0.52 (0.51, 0.53) | 0.76 (0.75, 0.77) | **0.52 (0.50, 0.53)** | 0.74 (0.73, 0.76) |
| Integrated Gradients | 0.55 (0.54, 0.56) | 0.74 (0.73, 0.75) | 0.65 (0.64, 0.67) | 0.73 (0.71, 0.74) |
| SmoothGrad | 0.70 (0.69, 0.71) | 0.72 (0.71, 0.73) | 0.72 (0.71, 0.73) | 0.73 (0.72, 0.75) |
| DeepSHAP | 0.65 (0.64, 0.66) | 0.79 (0.78, 0.80) | 0.69 (0.68, 0.71) | 0.74 (0.73, 0.75) |
| CXPlain | 0.56 (0.55, 0.57) | 0.71 (0.70, 0.72) | 0.60 (0.58, 0.61) | 0.72 (0.71, 0.74) |

For Imagenette, GradCAM performs competitively with FastSHAP on Exclusion AUC and KernelSHAP-S marginally beats FastSHAP on Inclusion AUC. KernelSHAP-S also outperforms on Inclusion AUC with CIFAR-10, which is perhaps surprising given its high level of noise (fig. 4). However, KernelSHAP-S does not do as well when evaluated using Exclusion AUC, and GradCAM does not do as well on Inclusion AUC. The remaining methods are, by and large, not competitive on either metric (except DeepSHAP on CIFAR-10 Inclusion AUC). An accurate explanation should perform well on both metrics, so these results show that FastSHAP provides the most versatile explanations, because it is the only approach to excel at both Inclusion and Exclusion AUC.

Finally, we also test FastSHAP's robustness to limited training data. In appendix E, we find that FastSHAP outperforms most baseline methods on Inclusion and Exclusion AUC when using just 25% of the Imagenette data, and that it remains competitive when using just 10%.

## 6.5 SPEED EVALUATION

The image experiments were run using 8 cores of an Intel Xeon Gold 6148 processor and a single NVIDIA Tesla V100. Table 2 records the time required to explain 1,000 images. For Fast-SHAP, KernelSHAP-S and CXPlain, we also report the time required to train the surrogate and/or explainer models.

The amortized explanation methods, FastSHAP and CXPlain, incur a fixed training cost but very low marginal cost for each explanation. The gradient-based methods are slightly slower, but KernelSHAP requires significantly more time. These results suggest that FastSHAP is well suited for real-time applications where it is cru-

Table 2: **Training and explanation run-times for 1,000 images (in minutes).**

| | | CIFAR-10 | Imagenette |
|---|---|---|---|
| Explain | FastSHAP | **0.04** | **0.04** |
| | KernelSHAP | 453.69 | 1089.50 |
| | KernelSHAP-S | 460.10 | 586.12 |
| | GradCAM | 0.38 | 0.30 |
| | IntGrad | 0.91 | 0.92 |
| | SmoothGrad | 1.00 | 1.05 |
| | DeepSHAP | 5.39 | 6.01 |
| | CXPlain | **0.04** | **0.04** |
| Train | FastSHAP | 693.57 | 146.49 |
| | KernelSHAP-S | 362.03 | 73.22 |
| | CXPlain | 538.49 | 93.00 |

cial to keep explanation times as low as possible. Further, when users need to explain a large quantity of data, FastSHAP's low explanation cost can quickly compensate for its training time.

## 7 DISCUSSION

In this work, we introduced FastSHAP, a method for estimating Shapley values in a single forward pass using a learned explainer model. To enable efficient training, we sidestepped the need for a training set and derived a learning approach from the Shapley value's weighted least squares characterization. Our experiments demonstrate that FastSHAP can produce accurate Shapley value estimates while achieving a significant speedup over non-amortized approaches, as well as more accurate image explanations than popular gradient-based methods.

While Shapley values provide a strong theoretical grounding for model explanation, they have not been widely adopted for explaining large-scale models due to their high computational cost. Fast-SHAP can solve this problem, making fast and high-quality explanations possible in fields such as computer vision and natural language processing. By casting model explanation as a learning problem, FastSHAP stands to benefit as the state of deep learning advances, and it opens a new direction of research for efficient Shapley value estimation.

## 8 REPRODUCIBILITY

Code to implement FastSHAP is available online in two separate repositories: `https://github.com/iancovert/fastshap` contains a PyTorch implementation and `https://github.com/neiljethani/fastshap/` a TensorFlow implementation, both with examples of tabular and image data experiments. The complete code for our experiments is available at `https://github.com/iclr1814/fastshap`, and details are described throughout section 5 and section 6, with model architectures and hyperparameters reported in appendix D. Proofs for our theoretical claims are provided in appendix A, appendix B, and appendix C.

## 9 ACKNOWLEDGEMENTS

We thank the reviewers for their thoughtful feedback, and we thank the Lee Lab for helpful discussions. Neil Jethani was partially supported by NIH T32 GM136573. Mukund Sudarshan was partially supported by a PhRMA Foundation Predoctoral Fellowship. Mukund Sudarshan and Rajesh Ranganath were partly supported by NIH/NHLBI Award R01HL148248, and by NSF Award 1922658 NRT-HDR: FUTURE Foundations, Translation, and Responsibility for Data Science. Ian Covert and Su-In Lee were supported by the NSF Awards CAREER DBI-1552309 and DBI-1759487; the NIH Awards R35GM128638 and R01NIAAG061132; and the American Cancer Society Award 127332-RSG-15-097-01-TBG.

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

## A   FASTSHAP GLOBAL OPTIMIZER

Here, we prove that FastSHAP is trained using an objective function whose global optimizer outputs the true Shapley values. Recall that the loss function for the explainer model $\phi_{\text{fast}}(\mathbf{x}, \mathbf{y}; \theta)$ is

$$\mathcal{L}(\theta) = \mathop{\mathbb{E}}_{p(\mathbf{x})} \mathop{\mathbb{E}}_{\text{Unif}(\mathbf{y})} \mathop{\mathbb{E}}_{p(\mathbf{s})} \left[ \left( v_{\mathbf{x},\mathbf{y}}(\mathbf{s}) - v_{\mathbf{x},\mathbf{y}}(\mathbf{0}) - \mathbf{s}^\top \phi_{\text{fast}}(\mathbf{x}, \mathbf{y}; \theta) \right)^2 \right].$$

As mentioned in the main text, it is necessary to force the model to satisfy the Efficiency constraint, or the property that the predictions from $\phi_{\text{fast}}(\mathbf{x}, \mathbf{y}; \theta)$ satisfy

$$\mathbf{1}^\top \phi_{\text{fast}}(x, y; \theta) = v_{x,y}(\mathbf{1}) - v_{x,y}(\mathbf{0}) \quad \forall \, x \in \mathcal{X}, y \in \mathcal{Y}.$$

One option for guaranteeing the efficiency property is to adjust the model outputs using their additive efficient normalization (see section 3.1). Incorporating this constraint on the predictions, we can then view the loss function as an expectation across $(\mathbf{x}, \mathbf{y})$ and write the expected loss for each sample $(x, y)$ as a separate optimization problem over the variable $\phi_{x,y} \in \mathbb{R}^d$:

$$\min_{\phi_{x,y}} \mathop{\mathbb{E}}_{p(\mathbf{s})} \left[ \left( v_{x,y}(\mathbf{s}) - v_{x,y}(\mathbf{0}) - \mathbf{s}^\top \phi_{x,y} \right)^2 \right] \tag{9}$$
$$\text{s.t.} \quad \phi_{x,y} = v_{x,y}(\mathbf{1}) - v_{x,y}(\mathbf{0}).$$

This is a constrained weighted least squares problem with a unique global minimizer, and it is precisely the Shapley value's weighted least squares characterization (see eq. (3)). We can therefore conclude that the optimal prediction for each pair $(x, y)$ is the true Shapley values, or that $\phi_{x,y}^* = \phi(v_{x,y})$. As a result, the global optimizer for our objective is a model $\phi_{\text{fast}}(\mathbf{x}, \mathbf{y}; \theta^*)$ that outputs the true Shapley values almost everywhere in the data distribution $p(\mathbf{x}, \mathbf{y})$.

Achieving the global optimum requires the ability to sample from $p(\mathbf{x})$ (or a sufficiently large dataset), perfect optimization, and a function class for $\phi_{\text{fast}}(\mathbf{x}, \mathbf{y}; \theta)$ that is expressive enough to contain the global optimizer. The universal approximation theorem (Cybenko, 1989; Hornik, 1991) implies that a sufficiently large neural network can represent the Shapley value function to arbitrary accuracy as long as it is a continuous function. Specifically, we require the function $h_y(x) = \phi(v_{x,y})$ to be continuous in $x$ for all $y$. This holds in practice when we use a surrogate model parameterized by a continuous neural network, because $v_{x,y}(s)$ is continuous in $x$ for all $(s, y)$, and the Shapley value is a linear combination of $v_{x,y}(s)$ across different values of $s$ (see eq. (1)).

Another approach to enforce the efficiency property is by using efficiency regularization, or penalizing the efficiency gap in the explainer model's predictions (section 3.1). If we incorporate this regularization term with parameter $\gamma > 0$, then our objective yields the following optimization problem for each $(x, y)$ pair:

$$\min_{\phi_{x,y}} \mathop{\mathbb{E}}_{p(\mathbf{s})} \left[ \left( v_{x,y}(\mathbf{s}) - v_{x,y}(\mathbf{0}) - \mathbf{s}^\top \phi_{x,y} \right)^2 \right] + \gamma \left( v_{x,y}(\mathbf{1}) - v_{x,y}(\mathbf{0}) - \mathbf{1}^\top \phi_{x,y} \right)^2.$$

For finite hyperparameter values $\gamma \in [0, \infty)$, this problem relaxes the Shapley value's efficiency property and eliminates the requirement that predictions must sum to the grand coalition's value. However, as we let $\gamma \to \infty$, the penalty term becomes closer to the hard constraint in eq. (9). Note that in practice, we use finite values for $\gamma$ and observe sufficiently accurate results, whereas using excessively large $\gamma$ values would render gradient-based optimization ineffective.

## B   ADDITIVE EFFICIENT NORMALIZATION

Here, we provide a geometric interpretation for the additive efficient normalization step and prove that it is guaranteed to yield Shapley value estimates closer to their true values. Consider a game $v$ with Shapley values $\phi(v) \in \mathbb{R}^d$, and assume that we have Shapley values estimates $\hat{\phi} \in \mathbb{R}^d$ that do not satisfy the efficiency property. To force this property to hold, we can project these estimates

onto the *efficient hyperplane*, or the subset of $\mathbb{R}^d$ where the efficiency property is satisfied. This corresponds to solving the following optimization problem over $\phi_{\text{eff}} \in \mathbb{R}^d$:

$$\min_{\phi_{\text{eff}}} \ ||\phi_{\text{eff}} - \hat{\phi}||^2 \quad \text{s.t.} \quad \mathbf{1}^\top \phi_{\text{eff}} = v(\mathbf{1}) - v(\mathbf{0}).$$

We can solve the problem via its Lagrangian, denoted by $\mathcal{L}(\phi_{\text{eff}}, \nu)$, with the Lagrange multiplier $\nu \in \mathbb{R}$ as follows:

$$\mathcal{L}(\phi_{\text{eff}}, \nu) = ||\phi_{\text{eff}} - \hat{\phi}||^2 + \nu\Big(v(\mathbf{1}) - v(\mathbf{0}) - \mathbf{1}^\top \phi_{\text{eff}}\Big)$$

$$\Rightarrow \phi_{\text{eff}}^* = \hat{\phi} - \mathbf{1}\frac{v(\mathbf{1}) - v(\mathbf{0}) - \mathbf{1}^\top \hat{\phi}}{d}.$$

This transformation, where the efficiency gap is split evenly and added to each estimate, is known as *additive efficient normalization* (Ruiz et al., 1998). We implement it as an output layer for Fast-SHAP's predictions to ensure that they satisfy the efficiency property (section 3). This step can therefore be understood as a projection of the network's output onto the efficient hyperplane.

The normalization step is guaranteed to produce corrected estimates $\phi_{\text{eff}}^*$ that are closer to the true Shapley values $\phi(v)$ than the original estimates $\hat{\phi}$. To see this, note that the projection step guarantees that $\hat{\phi} - \phi_{\text{eff}}^*$ and $\phi_{\text{eff}}^* - \phi(v)$ are orthogonal vectors, so the Pythagorean theorem yields the following inequality:

$$||\phi(v) - \hat{\phi}||^2 = ||\phi(v) - \phi_{\text{eff}}^*||^2 + ||\phi_{\text{eff}}^* - \hat{\phi}||^2$$
$$\geq ||\phi(v) - \phi_{\text{eff}}^*||^2.$$

## C  REDUCING GRADIENT VARIANCE

Recall that our objective function $\mathcal{L}(\theta)$ is defined as follows:

$$\mathcal{L}(\theta) = \mathop{\mathbb{E}}_{p(\mathbf{x})} \mathop{\mathbb{E}}_{\text{Unif}(\mathbf{y})} \mathop{\mathbb{E}}_{p(\mathbf{s})} \Big[ \big(v_{\mathbf{x},\mathbf{y}}(\mathbf{s}) - v_{x,y}(\mathbf{0}) - \mathbf{s}^\top \phi_{\text{fast}}(\mathbf{x}, \mathbf{y}; \theta)\big)^2 \Big].$$

The objective's gradient is given by

$$\nabla_\theta \mathcal{L}(\theta) = \mathop{\mathbb{E}}_{p(\mathbf{x})} \mathop{\mathbb{E}}_{\text{Unif}(\mathbf{y})} \mathop{\mathbb{E}}_{p(\mathbf{s})} \Big[ \nabla(\mathbf{x}, \mathbf{y}, \mathbf{s}; \theta) \Big], \tag{10}$$

where we define

$$\nabla(x, y, s; \theta) := \nabla_\theta \big(v_{x,y}(s) - v_{x,y}(\mathbf{0}) - s^\top \phi_{\text{fast}}(x, y; \theta)\big)^2.$$

When FastSHAP is trained with a single sample $(x, y, s)$, the gradient covariance is given by $\text{Cov}\big(\nabla(\mathbf{x}, \mathbf{y}, \mathbf{s}; \theta)\big)$, which may be too large for effective optimization. We use several strategies to reduce gradient variance. First, given a model that outputs estimates for all classes $y \in \{1, \dots, K\}$, we calculate the loss jointly for all classes. This yields gradients that we denote as $\nabla(\mathbf{x}, \mathbf{s}; \theta)$, defined as

$$\nabla(x, s; \theta) := \mathop{\mathbb{E}}_{\text{Unif}(\mathbf{y})} [\nabla(x, \mathbf{y}, s; \theta)],$$

where we have the relationship

$$\text{Cov}\big(\nabla(\mathbf{x}, \mathbf{s}; \theta)\big) \preceq \text{Cov}\big(\nabla(\mathbf{x}, \mathbf{y}, \mathbf{s}; \theta)\big)$$

due to the law of total covariance. Next, we consider minibatches of $b$ independent $x$ samples, which yields gradients $\nabla_b(\mathbf{x}, \mathbf{s}; \theta)$ with covariance given by

$$\text{Cov}\big(\nabla_b(\mathbf{x}, \mathbf{s}; \theta)\big) = \frac{1}{b}\text{Cov}\big(\nabla(\mathbf{x}, \mathbf{s}; \theta)\big).$$

We then consider sampling $m$ independent coalitions $s$ for each input $x$, resulting in the gradients $\nabla_b^m(\mathbf{x}, \mathbf{s}; \theta)$ with covariance given by

$$\text{Cov}\big(\nabla_b^m(\mathbf{x}, \mathbf{s}; \theta)\big) = \frac{1}{mb}\text{Cov}\big(\nabla(\mathbf{x}, \mathbf{s}; \theta)\big).$$

Finally, we consider a *paired sampling* approach, where each sample $s \sim p(\mathbf{s})$ is paired with its complement $\mathbf{1} - s$. Paired sampling has been shown to reduce KernelSHAP's variance (Covert and Lee, 2021), and our experiments show that it helps improve FastSHAP's accuracy (appendix D).

The training algorithm in the main text is simplified by omitting these gradient variance reduction techniques, so we also provide algorithm 2 below, which includes minibatching, multiple coalition samples, paired sampling, efficiency regularization and parallelization over all output classes.

---

**Algorithm 2:** Full FastSHAP training

---

**Input:** Value function $v_{x,y}$, learning rate $\alpha$, batch size $b$, samples $m$, penalty parameter $\gamma$
**Output:** FastSHAP explainer $\phi_{\text{fast}}(\mathbf{x}, \mathbf{y}; \theta)$
initialize $\phi_{\text{fast}}(\mathbf{x}, \mathbf{y}; \theta)$
**while** *not converged* **do**
    set $\mathcal{R} \leftarrow 0$, $\mathcal{L} \leftarrow 0$
    **for** $i = 1, \ldots, b$ **do**
        sample $x \sim p(\mathbf{x})$
        **for** $y = 1, \ldots, K$ **do**
            predict $\hat{\phi} \leftarrow \phi_{\text{fast}}(x, y; \theta)$
            calculate $\mathcal{R} \leftarrow \mathcal{R} + \left(v_{x,y}(\mathbf{1}) - v_{x,y}(\mathbf{0}) - \mathbf{1}^\top\hat{\phi}\right)^2$    // Pre-normalization
            **if** *normalize* **then**
                set $\hat{\phi} \leftarrow \hat{\phi} + d^{-1}\left(v_{x,y}(\mathbf{1}) - v_{x,y}(\mathbf{0}) - \mathbf{1}^\top\hat{\phi}\right)$
            **end**
            **for** $j = 1, \ldots, m$ **do**
                **if** *paired sampling* and $i \bmod 2 = 0$ **then**
                    set $s \leftarrow \mathbf{1} - s$   // Invert previous subset
                **else**
                    sample $s \sim p(\mathbf{s})$
                **end**
                calculate $\mathcal{L} \leftarrow \mathcal{L} + \left(v_{x,y}(s) - v_{x,y}(\mathbf{0}) - s^\top\hat{\phi}\right)^2$
            **end**
        **end**
    **end**
    update $\theta \leftarrow \theta - \alpha\nabla_\theta\left(\frac{\mathcal{L}}{bmK} + \gamma\frac{\mathcal{R}}{bK}\right)$
**end**

---

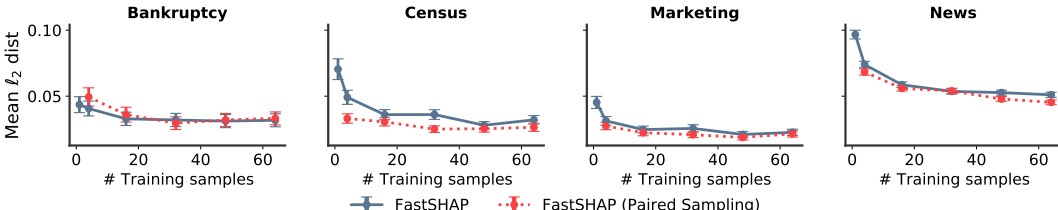

Figure 6: **FastSHAP accuracy as a function of the number of training samples.** The results show that using more **s** samples per **x** improves FastSHAP's closeness to the ground truth Shapley values, as does the use of paired sampling.

## D    FastSHAP MODELS AND HYPERPARAMETERS

In this section, we describe the models and architectures used for each dataset, as well as the hyperparameters used when training FastSHAP.

### D.1    MODELS

**Tabular datasets.**  For the original model $f(\mathbf{x}; \eta)$, we use neural networks for the `news` and `marketing` datasets and gradient boosted trees for the `census` (LightGBM (Ke et al., 2017)) and `bankruptcy` (XGBoost (Chen and Guestrin, 2016)) datasets. The FastSHAP model $\phi_{\text{fast}}(\mathbf{x}, \mathbf{y}; \theta)$ and the surrogate model $p_{\text{surr}}(\mathbf{y} \mid m(\mathbf{x}, \mathbf{s}); \beta)$ are implemented using neural networks that consist of 2-3 fully connected layers with 128 units and ReLU activations. The $p_{\text{surr}}$ models use a softmax output layer, while $\phi_{\text{fast}}$ has no output activation. The models are trained using Adam with a learning rate of $10^{-3}$, and we use a learning rate scheduler that multiplies the learning rate by a factor of $0.5$ after 3 epochs of no validation loss improvement. Early stopping was triggered after the validation loss ceased to improve for 10 epochs.

**Image datasets.**  The models $f(\mathbf{x}; \eta)$ and $p_{\text{surr}}$ are ResNet-50 models pretrained on Imagenet. We use these without modification to the architecture and fine-tune them on each image dataset. To create the $\phi_{\text{fast}}(\mathbf{x}, \mathbf{y}; \theta)$ model, we modify the architecture to return a tensor of size $14 \times 14 \times K$. First, the layers after the 4th convolutional block are removed; the output of this block is $14 \times 14 \times 256$. We then append a 2D convolutional layer with $K$ filters, each of size $1 \times 1$, so that the output is $14 \times 14 \times K$ and the $y$th $14 \times 14$ slice corresponds to the superpixel-level Shapley values for each class $y \in \mathcal{Y}$. Each model is trained using Adam with a learning rate of $10^{-3}$, and we use a learning rate scheduler that multiplies the learning rate by a factor of $0.8$ after 3 epochs of no validation loss improvement. Early stopping was triggered after the validation loss ceased to improve for 20 epochs.

### D.2    FastSHAP HYPERPARAMETERS

We now explore various settings of FastSHAP's hyperparameters and observe their impact on FastSHAP's performance. There are two types of hyperparameters: sampling hyperparameters, which affect the number of samples of **s** taken during training, and efficiency hyperparameters, which control how we enforce the Efficiency constraint. Sampling hyperparameters include: (1) whether to use paired sampling, and (2) the number of samples of **s** per **x** to take during training. Efficiency hyperparameters include: (1) the choice of $\gamma$ in eq. (4), and (2) whether to perform the additive efficient normalization during training, inference or both.

To understand the effect of sampling hyperparameters, we perform experiments using the same tabular datasets from the main text. We use the in-distribution value function $p_{\text{surr}}$ and compute the ground truth SHAP values the same way as in our previous experiments (i.e., by running KernelSHAP to convergence).

Figure 6 shows the mean $\ell_2$ distance between FastSHAP's estimates and the ground truth. We find that across all four datasets, increasing the number of training samples of **s** generally improves the mean $\ell_2$ distance to ground truth. We also find that for any fixed number of samples (greater than 1), using paired sampling improves FastSHAP's accuracy.

Table 3 shows the results of an ablation study for the efficiency hyperparameters. *Normalization* (or *Norm.*) refers to the additive efficient normalization step (applied during training and inference,

Table 3: **FastSHAP ablation results.** The distance to the ground truth Shapley values is displayed for several FastSHAP variations, showing that normalization helps and that the penalty is unnecessary.

|  | Census | | Bankruptcy | |
| --- | --- | --- | --- | --- |
|  | $\ell_2$ | $\ell_1$ | $\ell_2$ | $\ell_1$ |
| Normalization | **0.0229** | **0.0863** | **0.0295** | **0.2436** |
| Normalization + Penalty | 0.0261 | 0.0971 | 0.0320 | 0.2740 |
| Inference Norm. | 0.0406 | 0.1512 | 0.0407 | 0.3450 |
| Inference Norm. + Penalty | 0.0452 | 0.1671 | 0.0473 | 0.4471 |
| No Norm. | 0.0501 | 0.1933 | 0.0408 | 0.3474 |
| No Norm. + Penalty | 0.0513 | 0.1926 | 0.0474 | 0.4490 |

or only during inference), and *penalty* refers to the efficiency regularization technique with the parameter set to $\gamma = 0.1$. We find that using normalization during training uniformly achieves better results than without normalization or with normalization only during inference. The efficiency regularization approach proves to be less effective, generally leading to less accurate Shapley value estimates. Based on these results, we opt to use additive efficient normalization in our tabular data experiments.

# E    ADDITIONAL RESULTS FOR IMAGE EXPERIMENTS

In this section, we provide additional results for the FastSHAP image experiments.

## E.1    INCLUSION AND EXCLUSION METRICS

Table 4 shows our inclusion and exclusion metrics when replicated using log-odds rather than accuracy. Similar to our metrics described in the main text, we choose the class predicted by the original model for each image, and we measure the average log-odds for that class as we include or exclude important features according to the explanations generated by each method. The results confirm roughly the same ordering between methods, with FastSHAP being the only method to achieve strong results on both metrics for both datasets. Figure 7 shows the raw inclusion and exclusion curves for both the accuracy and log-odds-derived metrics.

Table 4: **Exclusion and Inclusion AUCs calculated using the average log-odds of the predicted class.**

|  | CIFAR-10 | | Imagenette | |
| --- | --- | --- | --- | --- |
|  | Exclusion AUC | Inclusion AUC | Exclusion AUC | Inclusion AUC |
| **FastSHAP** | **5.92 (5.62, 6.14)** | 5.36 (5.16, 5.63) | **7.98 (7.68, 8.33)** | 5.40 (5.16, 5.60) |
| KernelSHAP | 9.88 (9.55, 10.20) | 5.36 (5.14, 5.63) | 10.68 (10.36, 11.00) | 5.07 (4.81, 5.31) |
| KernelSHAP-S | 8.01 (7.68, 8.34) | **6.80 (6.65, 6.96)** | 9.39 (9.11, 9.66) | **6.01 (5.78, 6.26)** |
| GradCAM | 7.75 (7.44, 8.09) | 4.99 (4.81, 5.26) | 7.77 (7.49, 8.05) | 4.65 (4.40, 4.89) |
| Integrated Gradients | 8.34 (8.03, 8.61) | 4.58 (4.37, 4.85) | 10.14 (9.79, 10.46) | 4.34 (4.10, 4.58) |
| SmoothGrad | 10.99 (10.67, 11.29) | 4.30 (4.08, 4.58) | 11.19 (10.84, 11.48) | 4.47 (4.24, 4.70) |
| DeepSHAP | 9.96 (9.61, 10.24) | 5.47 (5.28, 5.76) | 10.93 (10.61, 11.20) | 4.63 (4.38, 4.85) |
| CXPlain | 8.34 (8.00, 8.58) | 4.02 (3.80, 4.31) | 9.13 (8.83, 9.41) | 4.33 (4.11, 4.57) |

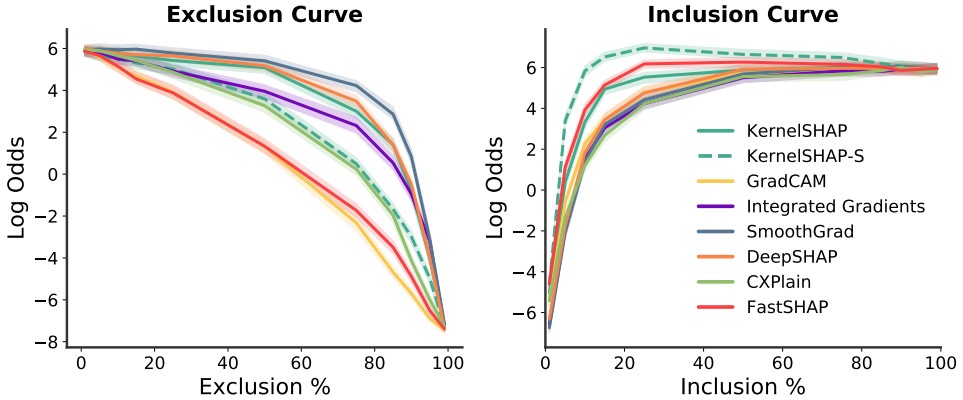

(a) **Imagenette: Log-odds derived inclusion and exclusion curves.**

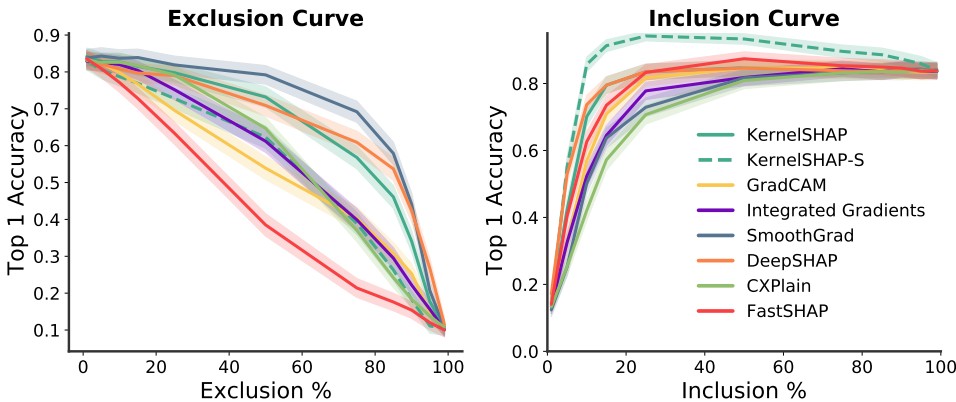

(b) **CIFAR-10: Accuracy derived inclusion and exclusion curves.**

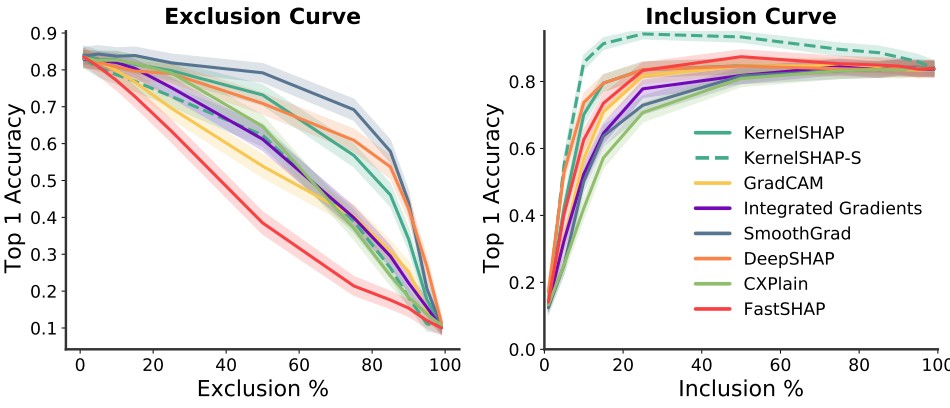

(c) **CIFAR-10: Log-odds derived inclusion and exclusion curves.**

Figure 7: **Additional inclusion and exclusion curves.** The change in top-1 accuracy or average log-odds of the predicted class as an increasing percentage of the pixels estimated to be important are excluded (left) or included (right) from the set of 1,000 images.

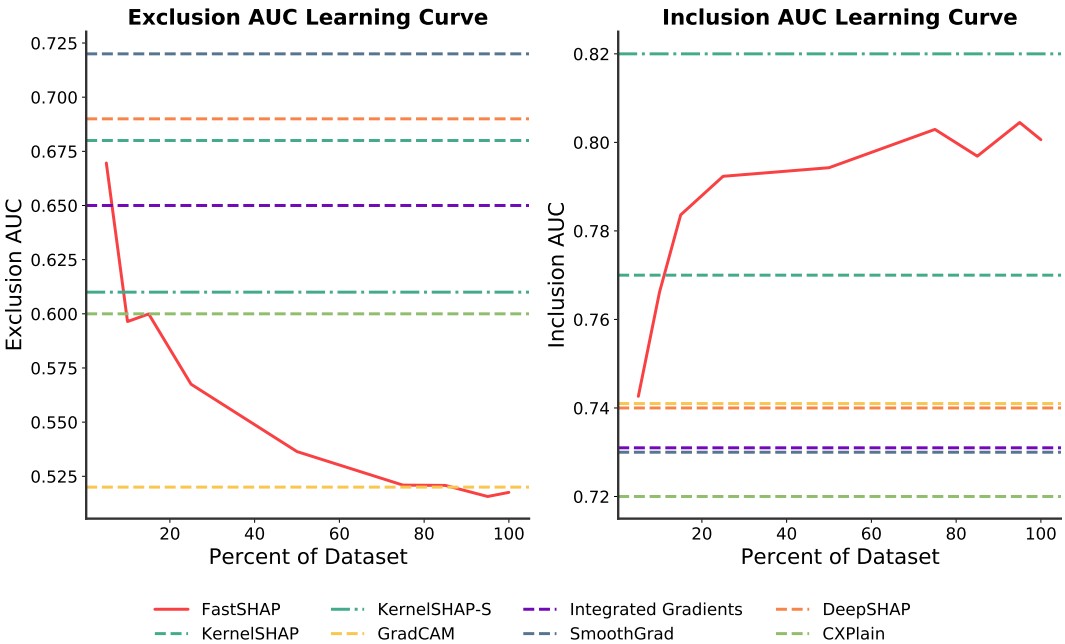

Figure 8: **FastSHAP robustness to limited data.** The curves are generated by training FastSHAP with varying portions of the Imagenette dataset and evaluating the Inclusion and Exclusion AUC. Horizontal lines show the Exclusion and Inclusion AUCs for each of the baseline methods, as reported in table 1.

## E.2    FASTSHAP ROBUSTNESS TO LIMITED DATA

To test FastSHAP's robustness to the size of the training data, we compare its performance when trained with varying amounts of the Imagenette dataset. Figure 8 plots the change in inclusion and exclusion AUC, calculated using top-1 accuracy, achieved when training FastSHAP with 95%, 85%, 75%, 50%, 25%, 15%, 10%, and 5% of the training dataset. We find that FastSHAP remains competitive when using just 10% of the data, and that it outperforms most baseline methods by a large margin when using just 25%.

## E.3    EXAMPLE FASTSHAP IMAGE EXPLANATIONS

Finally, we show additional explanations generated by FastSHAP and the baseline methods for both CIFAR-10 and Imagenette.

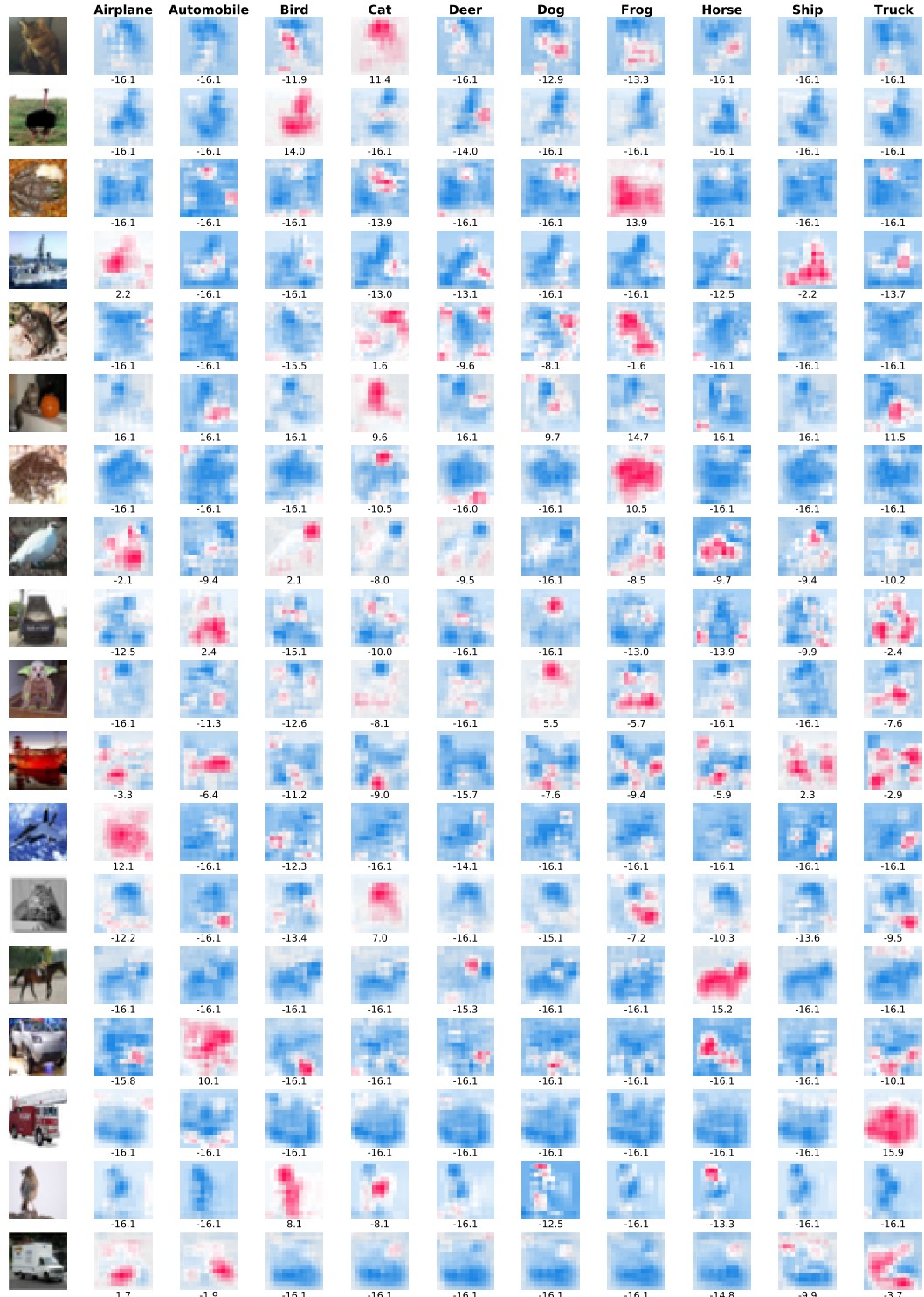

Figure 9: **Explanations generated by FastSHAP for 18 randomly selected CIFAR-10 images.** Each column corresponds to a CIFAR-10 class, and the model's prediction (in logits) is provided below each image.

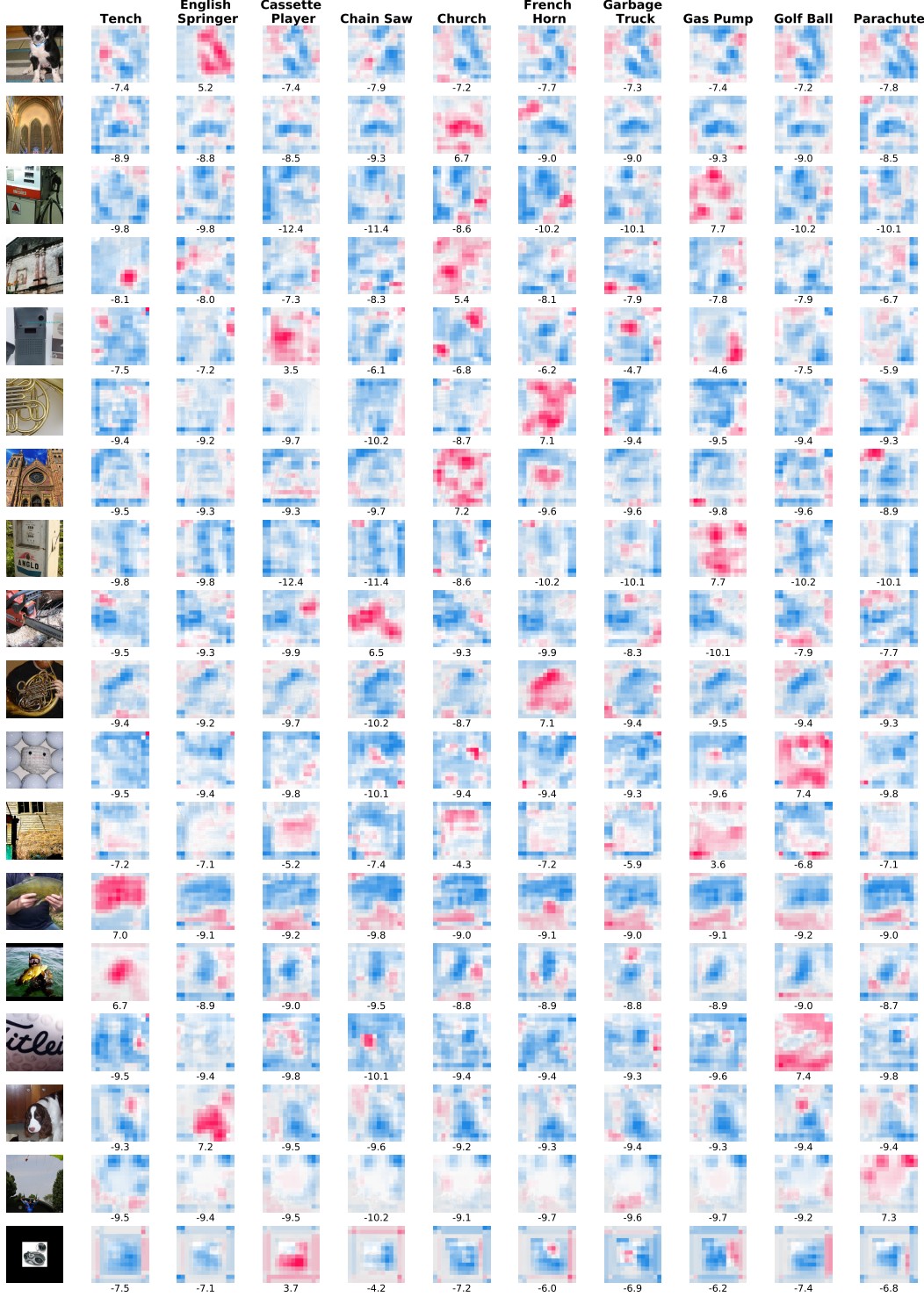

Figure 10: **Explanations generated by FastSHAP for 18 randomly selected Imagenette images.** Each column corresponds to an Imagenette class, and the model's prediction (in logits) is provided below each image.

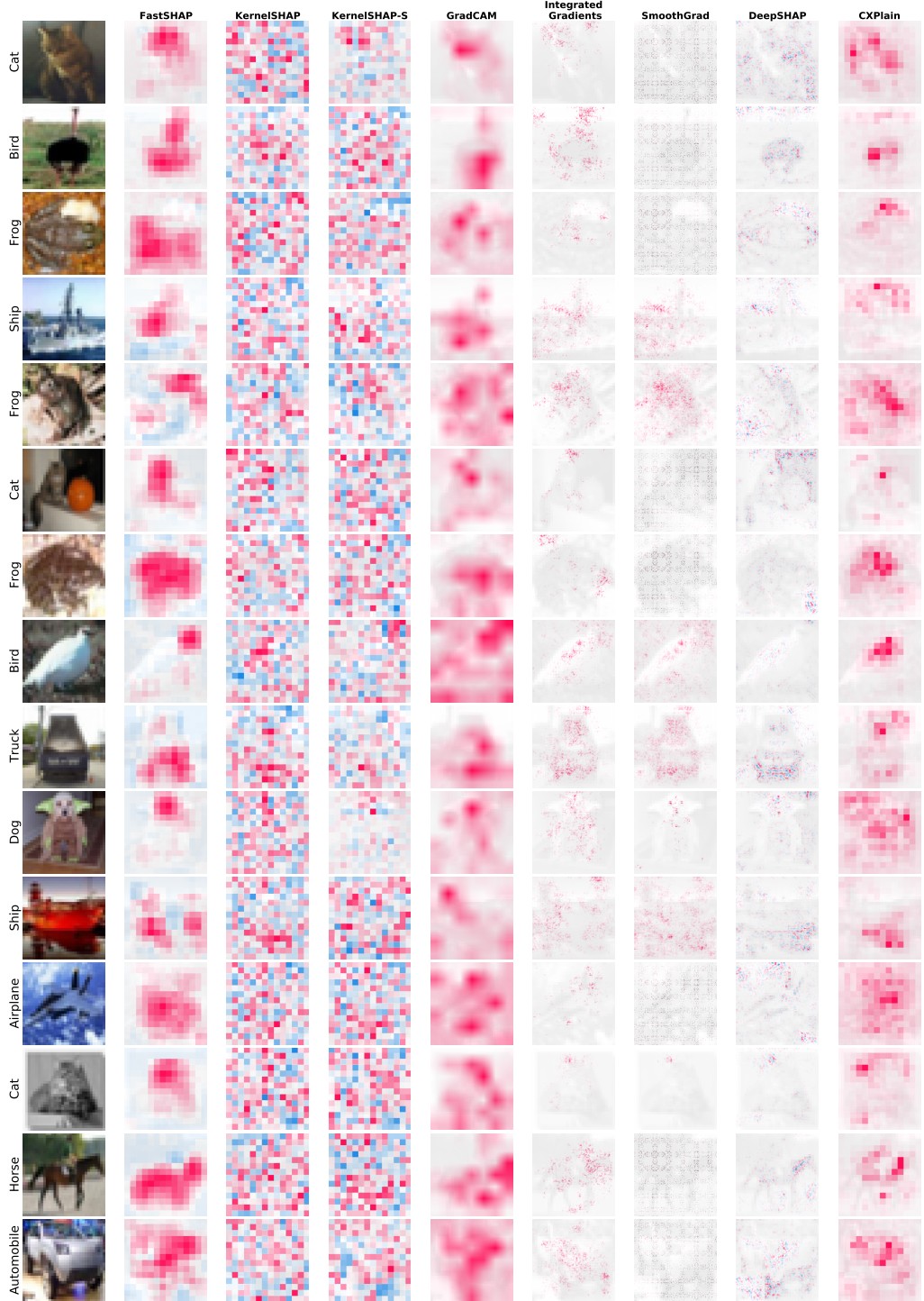

Figure 11: **Explanations generated for the predicted class for 15 randomly selected CIFAR-10 images.** Each column corresponds to an explanation method, and each row is labeled with the image's corresponding class.

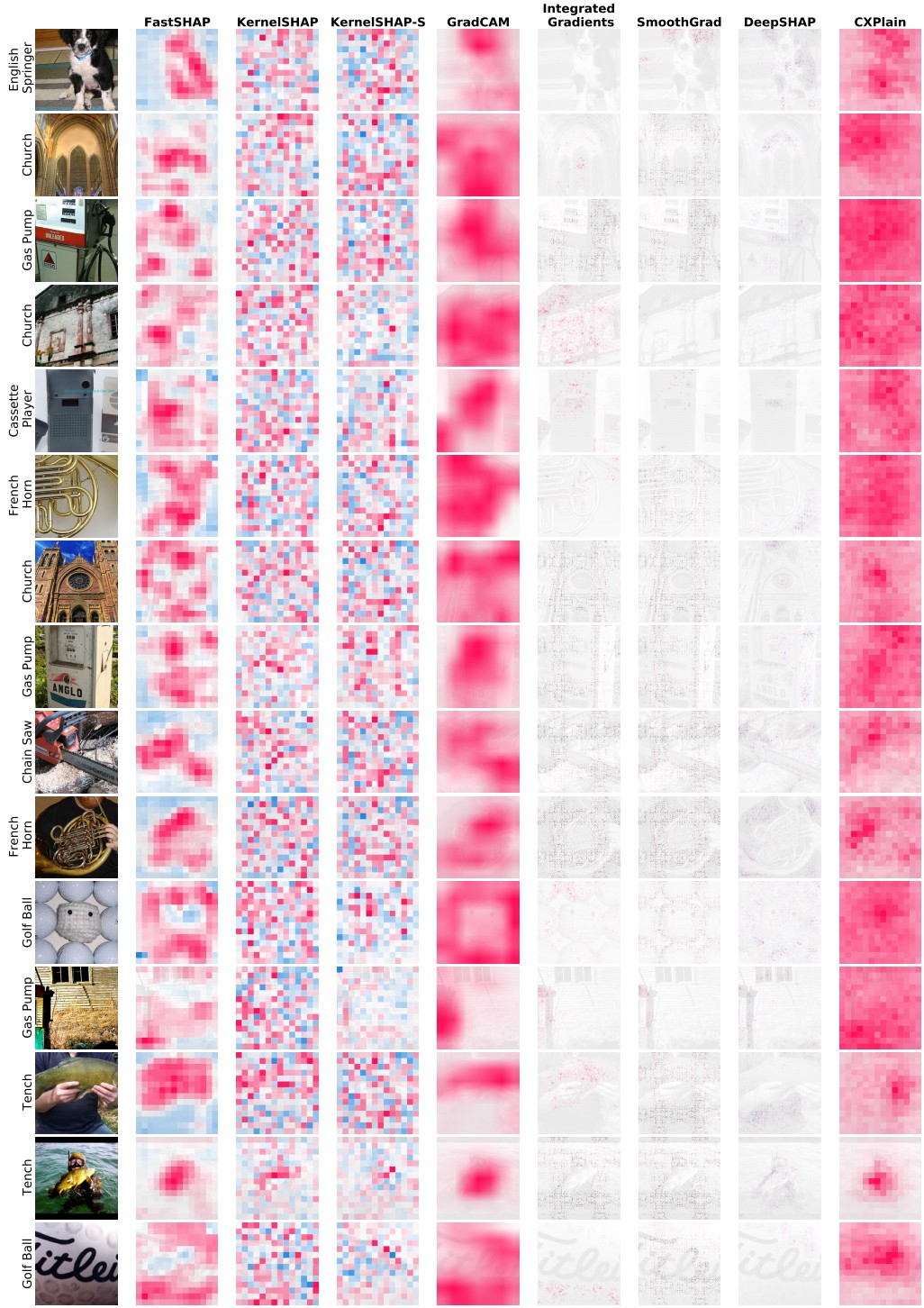

Figure 12: **Explanations generated for the predicted class for 15 randomly selected Imagenette images.** Each column corresponds to an explanation method, and each row is labeled with the image's corresponding class.

