# OpenReview forum: "FastSHAP: Real-Time Shapley Value Estimation"
_ICLR.cc/2022/Conference — ICLR 2022 Poster_

### Official Review · Reviewer_zqFs · 2021-10-20

**Correctness:** 3
**Technical Novelty And Significance:** 3
**Empirical Novelty And Significance:** 3
**Recommendation:** 5
**Confidence:** 3

**Main Review:**

Post-rebuttal:
The response partly addresses my concerns. However, I think the organization needs to be improved. The core contribution should be highlighted and the connection between sections should be strenthened.
-------------------------------------------------
Pros:
1. Efficiently estimating the Shapley value without groundtruth label is highly demanded, and the motivation is very clearly and in high significance.

2. The experimental results including the exclusion and inclusion curve show the approximation accuracy of the proposed method. Meanwhile, the runtime is competitive with the SOTA method.

Cons:
1. The main problem is the writing, and the organization of this paper should be improved. Since the presented method is based on the prior work KernelSHAP, the detailed information of KernelSHAP should be explained well. For example, what is $v_{x,y}(0)$ and $v_{x,y}(s)$? Meanwhile, the connection between Section 3.1 and 3.2 is not strong although the subtitle is connected.

2. The novelty is also concerned. Eq. (6)-(8) all comes from prior works and the authors seem to combine the results without obvious modification. I think the contribution of extending other theoretical results to the efficient Shapley value estimation should be clarified clearly.

3. The discussion for the experimental results are not sufficient. For example, FastSHAP outperforms all listed baselines in the exclusion curve but underforms KernelSHAP-S in the inclusion curve. What is the reason for the counter-intuitive phenomenon? Meanwhile, the bold results in Table 1 is confusing.

**Summary Of The Paper:**

This paper proposes FastSHAP to efficiently estimate the Shapley value in a single forward pass using a learned explainer model. Since there is no label to train the Shapley value estimator, stochastic gradient optimization using a weighted least squares-like objective function is applied to train FastSHAP. The experimental results and the deployment efficiency shows the superiority of the presented method.



**Summary Of The Review:**

Please see the Cons in the main review.

---

> ### Author Response · Authors · 2021-11-16
> **Response to reviewer zqFs**
>
> We'd like to thank the reviewer for their feedback. Please feel free to post responses for further clarifications.
>
> - We are surprised to hear the reviewer’s concerns about the clarity of our writing, the other reviewers did not point out this issue. We have space constraints and cannot provide a much more detailed review of the relevant background material, but we would like to point out that Section 2 is dedicated to background information on Shapley values and Section 2.2 is specifically dedicated to KernelSHAP. For more information, we can direct readers to the relevant papers to learn more (Lundberg & Lee 2017 and Covert & Lee 2020).
> - We have added a phrase to explain more clearly what $v_{x, y}$ represents: it shows how the prediction for class $y$ varies as different features are removed from $x$.
> - To clarify what is novel about this work: it’s the idea of amortizing Shapley value computation across multiple examples using a learned explainer model. Our core contribution is the loss function and training routine, which is described in section 3.1. Section 3.2 provides the remaining information necessary to understand how our models are implemented, and we can understand the desire to separate this part, but we think this presentation is preferable because it focuses the method’s description in a single section.
> - On the question of which equations represent new contributions: the most important equations are eqs. 4-6. Eq. 4 shows our custom loss function for training FastSHAP and eq. 5 represents our analysis of its optimality condition. Eq. 6 (the additive efficient normalization) has been discussed in some prior work, but never in the context of training deep learning models with an efficiency constraint.
> - It’s true that KernelSHAP performs slightly better with the Inclusion AUC score: it performs best while FastSHAP is in second place (or tied for first or second). However, it’s important to notice that KernelSHAP does much worse with the Exclusion AUC score. To explain what this means, it suggests that when KernelSHAP-S assigns features low attribution scores, those features still contain information that pushes the prediction towards the target class (which is not a desirable property). Both metrics (Inclusion and Exclusion AUC) are important for an explanation’s accuracy, and only FastSHAP does well on both metrics.
> - In Table 1, the best method in each column is bolded, and any method that ties due to 95% confidence intervals is also bolded. We have clarified this in the table caption.

---

> ### Author Response · Authors · 2021-11-27
> **Review Updates**
>
> Again, thank you for your feedback. We hope that our response has clarified the questions and concerns in your review, and we would greatly appreciate it if you could provide us with any additional feedback. We are happy to address any reservations you may have regarding FastSHAP or its presentation. If our response has been satisfactory, we would love it if you would consider raising your score.

---

### Official Review · Reviewer_Asf6 · 2021-10-27

**Correctness:** 3
**Technical Novelty And Significance:** 3
**Empirical Novelty And Significance:** 3
**Recommendation:** 6
**Confidence:** 4

**Main Review:**

The FastSHAP method is elegant and provides an effective solution to the computational bottleneck of Shapley value estimation. However, there are concerns and questions below about the paper:

- Why is the loss function in L2 form? Why don't you use other norm functions?

- How can one pick the learning rate of the method? What type of validation reward would be used?

- It is unclear how to optimally pick the loss coefficient for the efficiency gap objective.

- What are the approximations that the optimal solution for Eq. (7) is given by Eq. (8)? It should be explained better.

- In Fig. 4, the competitor models have very poor explanation quality, compared to the original papers that proposed them. How did you optimize their hyperparameters? Could there be unfairness in regenerating the results by those methods? Also, why doesn't it show results with the exact Shapley values?

- The gap with KernelSHAP is still somehow noticeable in Fig. 5 that reduces the impact of the contribution.

- Parallelization constitutes an important aspect of the runtime, but there is not much discussion on it. How can one parallelize different algorithmic operations, especially for the GPU implementation?

- What is the impact of dataset size on the accuracy of explanations? Some discussions on it would be needed.

**Summary Of The Paper:**

For fast estimation of Shapley values, a new method is proposed, based on a single forward pass using a learned explainer model. On tabular and image datasets, accurate and fast estimation of feature attributions are demonstrated.

**Summary Of The Review:**

Overall, the paper makes an important contribution for post-hoc explanation of ML models, in computationally efficient ways. The paper is well written with strong results, but there is further room for improvement.

---

> ### Author Response · Authors · 2021-11-16
> **Response to reviewer Asf6**
>
> We'd like to thank the reviewer for their constructive feedback. Please feel free to post responses for further clarifications. Below we address each point brought up in the review and our thoughts on each one.
>
> - The loss function is chosen so that the global optimizer is a function that outputs the true Shapley values. This property would not hold if we swapped the MSE component of the loss for an arbitrary norm (please see the proof for this property in appendix A, or a proof for the Shapley value’s weighted least squares characterization in Lundberg & Lee 2017).
> - To measure the model’s validation performance during training, we pre-computed a fixed validation set of $(x, y, s)$ tuples and calculated their loss. This validation loss can be used to select the learning rate, as well as any other training hyperparameters (e.g., the model architecture).
> - The parameter for the efficiency gap is not very important in practice. If the efficiency property is crucial, we recommend using the additive efficient normalization as in the tabular data experiments. Otherwise, we find in practice that $\gamma$ can be set to zero and have little effect on the outcome - see the computer vision experiments (although theoretically, this means that the global optimizer no longer outputs the exact Shapley values).
> - To clarify eqs. 7 and 8: eq. 7 gives a training objective for the surrogate model, and the global optimizer of this training objective is a model defined according to eq. 8. In practice, we only have access to our trained model, but it’s helpful to understand that this model is an approximation of marginalizing features out according to their conditional distribution. For more details, we recommend checking the cited papers that first used such an approach.
> - For the baseline methods, we used the default hyperparameters in all cases, and we are not sure that these results are unusual for these methods. Note that prior work, such as the ROAR paper (Hooker et al., 2019), have also found that IG and SmoothGrad perform poorly. To be more specific about the hyperparameters: 1) KernelSHAP’s only parameter here is the number of samples, and we used the default number from the shap implementation. 2) GradCAM has no tunable hyperparameters. 3) IG’s only hyperparameter is the number of points to use along the straight path, and we used the default number from the tf_explain implementation. 3) SmoothGrad’s only hyperparameter is the number of samples, and we used the default number from the tf_explain implementation. 4) DeepSHAP has no hyperparameters. 5) CXPlain requires choosing a model architecture, and we used the same model as FastSHAP: a ResNet50 modified to output explanations of the desired size.
> - It’s true that KernelSHAP performs slightly better with the Inclusion AUC score: it performs best while FastSHAP is in second place (or tied for first or second). However, notice that KernelSHAP does much worse with the Exclusion AUC score. Both metrics are important for an explanation’s accuracy, and only FastSHAP does well on both metrics.
> - We are not certain what the reviewer means regarding parallelization, so we can offer two answers. First, in the sense of parallelizing across multiple examples (which we recommend doing for variance reduction), this simply requires making a minibatch of predictions and summing the loss over all classes (which we recommend outputting simultaneously as different output dimensions). Alternatively, if the question is about training on multiple GPUs, then note that FastSHAP can be trained in a data-parallel setup just as easily as other deep learning models trained with SGD. We did not explore this, but it would offer a way to train FastSHAP even faster.
> - In general, a large dataset is helpful for the explainer model to learn the Shapley value function accurately and generalize to unseen examples. Our experiments used relatively large datasets where this should not be an issue, but it’s possible that FastSHAP would face limitations in small-dataset scenarios. To look into this question more carefully, we ran an experiment where FastSHAP was trained with different portions of the Imagenette dataset (note that this is the smaller of the two image datasets). We’ve added a [figure](https://imgur.com/a/1UTb9uC) summarizing the results to the supplement (appendix E): we find that the Inclusion and Exclusion AUC degrade when using less data, but not dramatically, and in particular, FastSHAP continues to outperform most of the baselines when using just 25% of the data.

---

> ### Author Response · Authors · 2021-11-27
> **Review Updates**
>
> Thank you again for your thoughtful feedback and overall positive review of our work! We hope that our response and new experiments (appendix E) have addressed any reservations you had about FastSHAP. Please let us know if you have any additional questions or concerns, we would be more than happy to address them. If our response has been satisfactory, then we would greatly appreciate it if you would consider updating your score.

---

### Official Review · Reviewer_GNuc · 2021-10-31

**Correctness:** 3
**Technical Novelty And Significance:** 3
**Empirical Novelty And Significance:** 4
**Recommendation:** 8
**Confidence:** 3

**Main Review:**

This paper proposes a significant idea of reducing the runtime for estimating the Shapley value. I really love the extensive simulations that the authors conducted.

Here are some comments:
1. ("... how to remove features") on page 2. I don't think that the goal of the works of (Aas et al., 2019, Janzing et al., 2020, etc.) is to remove features. Instead, these papers aim to choose the set function such that the resultant Shapley values have causal interpretation when there is a feature dependence.

2. In Eq. (4), Unif(y) seems undefined.

3. I think the paper should explicitly compare the computational complexity of the existing approximation methods, including the permutation-based algorithm [Štrumbelj and Kononenko, 2014], kernel-SHAP [Lundberg and Lee, 2017], etc., to see how the orders-of-magnitude speedup was achieved.

4. I am not sure what the authors want to tell from the simulation results in Figs. (2,3). I think the authors intend to tell that FastSHAP has higher performances compared to other works. However, I don't think this observation can be generalized because FastSHAP values are approximations of true Shapely Values onto the predefined function space. In other words, FastSHAP projects the solution of Eq. (3)  (i.e., the true Shapley) onto the predefined functional space. Then, we cannot make a general statement that FastSHAP achieves higher accuracy than other Shapley values because the functional space does not confine them.

5. For the practical benefit, I recommend adding the guideline for choosing the functional class for \phi_{fast}(x,y,\theta).

---
## Updated comments.

The authors addressed all the comments that I raised.

I agree that this paper provides a substantial empirical benefit for the XAI literature. However, I raised some issues on the novelty because I expected a theoretical guarantee for the outstanding performance of FastSHAP. Nevertheless, I decided to appreciate the paper's importance because it is evident that the proposed method can significantly reduce computational complexity.

I updated the scores toward acceptance because I believe the proposed approach will help the XAI community. Specifically, the scores are
* Technical Novelty And Significance: 2 -> 3
* Empirical Novelty And Significance: 2 -> 4
* Recommendation: 5 -> 8

### Some minor note
* Is "Single forward pass" broadly used words? I couldn't understand the contribution in the Introduction Section because "Single forward pass" looks exotic to me. I could grasp the contribution by seeing Eq. (4). It would be great if the contribution can be explained in simpler terms.



**Summary Of The Paper:**

This paper introduced FastSHAP, a new method for estimating Shapley values, reducing runtime significantly.

**Summary Of The Review:**

- Overall, I think there should be an explicit comparison of the computational complexities of FastSHAP and other algorithms.

- Instead of comparing the performance, a more extensive comparison of the runtime is needed.

---

> ### Author Response · Authors · 2021-11-16
> **Response to GNuc**
>
> We thank the reviewer for their positive feedback and questions, and we are happy to clarify several points below. Please feel free to post responses for further clarifications.
>
> - We use “removal” to refer to the general approach of discarding feature values from a model, please see Covert et al. (2020) for a large number of approaches that have been categorized as such. We recognize that some methods do this with the intention of modeling causal interventions; this is the case for Janzing et al. (2020), but not Aas et al. (2019), whose approach is not connected to causality. This is not an important distinction in our context because FastSHAP can work with any approach to removing/modifying/intervening on feature values.
> - We’ve clarified that $\text{Unif}(y)$ is a uniform distribution over the class variable $y$, thanks for pointing that out.
> - Comparing the computational complexity between algorithms is not feasible in this context. We could describe how the estimation error shrinks as a function of the number of samples for the baseline methods (these results are known in some cases but not others), but FastSHAP is much more difficult to characterize because it relies on a trained deep learning model. Such analysis is rarely performed in deep learning papers.
> - The results in Figures 2 and 3 are explained on p.6: to achieve the same $\ell_2$ distance to the ground truth Shapley values as FastSHAP, the baseline methods require 200-1000x more model evaluations. As the accuracy of deep learning approaches like FastSHAP is difficult to characterize theoretically, the point of this experiment is to understand empirically how accurate FastSHAP is. We’ve done this by measuring how long conventional methods would take to achieve the same accuracy. The exact result (200-1000x) depends on the dataset, as well as the model class and training hyperparameters, and the full details of our training setup are described in the supplement.
> - Using large neural networks is the right choice for FastSHAP for three reasons: 1) DNNs have a great degree of expressive capacity, 2) they offer a simple way to learn many-to-many mappings, and 3) they permit stochastic gradient optimization. These points are alluded to in the main text on p.3 (“given a large enough dataset and a sufficiently expressive model class for $\phi_{fast}$…”) and p.5 (under “Implementation details”), but we have added an additional note about this to section 5.
> - On the subject of run-time analysis, we hope the reviewer understands that FastSHAP is extremely fast for generating explanations: it requires only a single forward pass through the explainer model, whereas existing Shapley value estimators require hundreds or thousands of forward passes through the original model. To make this more concrete, please see our wall clock times for the computer vision experiments in Table 2: FastSHAP has a very low marginal cost for generating each explanation (particularly relative to KernelSHAP), and it can be faster in spite of its training time if there are enough samples to explain. The lower marginal cost also means that FastSHAP is much more practical than KernelSHAP for generating real-time explanations at inference/deployment time.

---

> > ### Comment · Reviewer_GNuc · 2021-11-25
> > **Response**
> >
> > Thank you for the detailed reply! Based on your response and after re-reading the paper, I decided to appreciate the paper's importance because it is evident that the proposed method can significantly reduce computational complexity. Therefore, I raised the overall scores.

---

> > > ### Author Response · Authors · 2021-11-27
> > > **Thank you!**
> > >
> > > We are happy to hear that you appreciate the novelty and significance of FastSHAP. Thank you for taking the time to engage with our response and re-read our paper. Cheers!

---

### Official Review · Reviewer_wDJz · 2021-11-02

**Correctness:** 3
**Technical Novelty And Significance:** 3
**Empirical Novelty And Significance:** Not applicable
**Recommendation:** 6
**Confidence:** 2

**Main Review:**

Strengths
- The paper proposes an effective method for estimating Shapley values that is much faster than existing works.
- The method is validated on structure and image datasets.
- The running time for different methods is well presented.

Weaknesses
- Besides the image classification tasks, could the proposed method work on other tasks, such as segmentation or NLP-related tasks?
- Compared with the FastSHAP, other methods can achieve better Exclusion AUC and Inclusion AUC for both CIFAR10 and Imagenette in Table 1. Could the authors explain more on why other methods can obtain better results?

**Summary Of The Paper:**

The paper works on improving the runtime for estimating Shapley values. The work introduces FastSHAP that estimates Shapley values with a learned explainer model. The method is validated on tabular and image datasets (CIFAR10 and Imagenette).

**Summary Of The Review:**

Overall, the paper introduces an interesting approach for estimating Shaley values in real run-time. The effectiveness of the method is well demonstrated across different tasks/datasets.

---

> ### Author Response · Authors · 2021-11-16
> **Response to wDJz**
>
> Thank you very much for the positive feedback on our work. To answer the questions raised in your review:
>
> - Yes, FastSHAP could prove quite useful for NLP tasks because that’s a domain with large models that are slow to evaluate and potentially many input features. We’re leaving that for future work, though.
>
> - To clarify, no method gets a better (lower) Exclusion AUC than FastSHAP on either dataset. As for the Inclusion AUC, KernelSHAP-S gets a marginally better (higher) score with Imagenette, and a more significantly larger score for CIFAR-10, but in both cases FastSHAP is tied for either first or second place. Notice, however, that KernelSHAP-S performs much worse with the Exclusion AUC on the two datasets. Both metrics are important for an explanation’s accuracy, and only FastSHAP performs well on both.

---

> ### Author Response · Authors · 2021-11-27
> **Review Updates**
>
> Thank you again for your positive review! We hope that our response has addressed any concerns you have about our work. Please let us know if you have other any additional questions, and we will be sure to address them. If our response has been satisfactory, then we would greatly appreciate it if you would consider updating your score.

---

### Author Response · Authors · 2021-11-16
**General response**

We thank the reviewers for providing helpful feedback on our manuscript and for their time. We are glad the reviewers appreciated that FastSHAP is capable of estimating Shapley values much faster than existing methods, that FastSHAP offers an elegant approach for training a model to estimate Shapley values without ground-truth labels, and that our experiments were thorough and well-presented. We have incorporated all the reviewers’ feedback by making several adjustments to the paper, and we hope that with these improvements the reviewers will consider raising their scores.

We respond to each of the reviewers’ comments individually, but there are several points we would like to highlight here. These are described below.

**Choice of squared error ($\ell_2$) in loss function.** One reviewer asked if our loss function could be adjusted to use different norms. The loss function is chosen so that the global optimizer is a function that outputs the true Shapley values, and this property would not hold if we swapped the MSE component of the loss for an arbitrary norm. For more details on this, please see the proof for this property in appendix A.

**FastSHAP evaluation with Inclusion and Exclusion AUC.** Some reviewers pointed out that KernelSHAP-S outperforms FastSHAP in some scenarios. To clarify, no method gets a better (lower) Exclusion AUC than FastSHAP on either dataset, but KernelSHAP-S does get a marginally better (higher) Inclusion AUC score with Imagenette and a more significantly larger score for CIFAR-10. (Note that in these cases FastSHAP is tied for either first or second place.) It is important to point out that KernelSHAP-S performs much worse with the Exclusion AUC on the two datasets, so it is only competitive in one of the two metrics. Both metrics are important for an explanation’s accuracy, and only FastSHAP performs well on both.

**Validation and hyperparameter tuning.** One reviewer asked about how we tune FastSHAP hyperparameters. We did this with a validation loss, which was calculated using a pre-computed (fixed) set of $(x, y, s)$ tuples. This validation loss can be used to select the learning rate, as well as any other training hyperparameters (e.g., the model architecture).

**Guidance for picking the model class for FastSHAP.** One reviewer asked what model class should be used for the FastSHAP explainer. Using deep neural networks (DNNs) is the best choice for three reasons: 1) large DNNs have a great degree of expressive capacity, 2) they offer a simple way to learn many-to-many mappings, and 3) they permit stochastic gradient optimization. These points were mentioned in the main text, but we have added an additional note about this to section 5. Using DNNs also means that we can use the best architecture for each data domain (e.g., MLPs for tabular data, CNNs for images, and transformers for text), and that FastSHAP’s performance should improve as these architectures continue to advance.

**Writing concerns.** We are surprised to see reviewer zqFs’s concerns about the clarity of our writing, but are glad to see that the other reviewers did not have this issue. We have space constraints and cannot provide a much more detailed review of the relevant background material, but we would like to point out that Section 2 is dedicated to background information on Shapley values and Section 2.2 is specifically dedicated to KernelSHAP. For more information on KernelSHAP, readers should see the cited papers (Lundberg & Lee 2017, Covert & Lee 2021). We’ve made a couple minor writing adjustments based on reviewer feedback, and we are open to other specific requests for clarifications.

---

### Author Response · Authors · 2021-12-07
**Reviewer discussion**

Dear reviewers,

Thank you very much for your valuable feedback that has helped us improve this work. We would also like to thank reviewers GNuc and zqFs for reading our rebuttal and updating their reviews. For the other two reviewers, wDJz and Asf6, we hope that we have addressed any concerns you may have had. If needed, we would be happy to provide any additional information to resolve your concerns. We would greatly appreciate it if you would read the rebuttal and consider updating your reviews.

Thanks very much,

The authors

---

### Decision · Program_Chairs · 2022-01-20

**Decision:**

Accept (Poster)

**Comment:**

The reviewers agree that the paper introduces an interesting approach for estimating Shaley values in real run-time. The effectiveness of the method is well demonstrated across different tasks/datasets.